# Macrophage-augmented intestinal organoids model virus-host interactions in enteric viral diseases and facilitate therapeutic development

Guige Xu[1,2,3,9], Jiangrong Zhou[2,9], Kuan Liu[2,4,9], Yining Wang[2], Theano Tsikari[5], Fang Qin[6], Francijna van den Hil [5], Patrick P. C. Boor[2], Ibrahim Ayada[2], Annemarie C. de Vries[2], Jiajing Li[2], Shijin Jiang[1], Dewy M. Offermans[2], Denis E. Kainov [7], Harry L. A. Janssen[2,8], Maikel P. Peppelenbosch [2], Marcel J. C. Bijvelds[2], Wenshi Wang[6], Valeria V. Orlova [5], Qiuwei Pan [2] & Pengfei Li [2,3] ✉

The pathogenesis of enteric viral infections is attributed to both viral replication and the resultant immune-inflammatory response. To recapitulate this complex pathophysiology, we engineer macrophage-augmented organoids (MaugOs) by integrating human macrophages into primary intestinal organoids. Echovirus 1, echovirus 6, rotavirus, seasonal coronavirus OC43 and SARS-CoV-2− known to directly invade the intestine− are used as disease modalities. We demonstrate that these viruses efficiently propagate in MaugOs and stimulate the host antiviral response. However, rotavirus, coronavirus OC43 and SARS-CoV-2, but not the two echoviruses, trigger inflammatory responses. Acetate, a microbial metabolite abundantly present in the intestine, potently inhibits virus-induced inflammatory responses in MaugOs, while differentially affecting viral replication in macrophages and organoids. Furthermore, we provide a proof-of-concept of combining antiviral agent with either anti-inflammatory regimen or acetate to simultaneously inhibit viral infection and inflammatory response in MaugOs. Collectively, these findings demonstrate that MaugOs are innovative tools for studying the complex virus-host interactions and advancing therapeutic development.

Enteric infections caused by many epidemic or endemic viruses result in major global health burdens including morbidity and mortality. Amongst various enteric viruses, rotavirus contributes to over two million hospitalizations each year and remains the leading cause of diarrhea-associated mortality in young children[1,2]. Enteroviruses comprise a large group of enteric viruses affecting millions of people globally[3]. Unlike rotavirus, enteroviruses (e.g., echovirus) rarely cause intestinal illness but typically result in clinical diseases only after

disseminating to other sites[4]. Non-enteric viruses such as pandemic SARS-CoV-2 and seasonal coronavirus OC43 primarily target the respiratory system[5], but can also invade the intestine and cause gastrointestinal manifestations[6].

Viral invasion often activates host innate immune response, with the interferon-mediated response serving as a universal, first-line defense mechanism against viral infection. This response leads to the transcription of hundreds of interferon-stimulated genes (ISGs)

to counteract the infection[7]. The inflammatory response involving immune cells, also acts as a host protective mechanism to eliminate the invading pathogens. However, this response must be timely terminated when no longer needed to prevent collateral tissue damage. Dysregulation of this feedback mechanism during severe acute viral infections, can result in pathological inflammation[8]. Furthermore, different types of viral pathogens orchestrate distinct features of virus-host interactions, which in turn shape the differential infection course and clinical outcomes in infected patients.

In enteric viral infections, severe clinical outcomes often involve localized or systemic inflammation, in addition to robust viral replication in the intestine[9]. Macrophages, in particular infiltrating monocytes differentiated macrophages, play a key role in orchestrating such pathological inflammation[10]. In response to pathogenic stimuli, macrophages can form a spectrum of inflammasomes, and the NLRP3 inflammasome is particularly relevant in recognizing viral infections[11]. The intestine harbors trillions of microorganisms, referring as microbiome, which produces a myriad of metabolites through fermentation[12]. Notably, specific gut metabolites are known to influence local and systemic viral infections, for example, through dampen or exacerbate pathological inflammation during infections[13].

Among patients with severe viral infections, viral propagation and the resulting inflammation concurrently contribute to mortality and morbidity. However, the first-line therapeutic approach for treating viral diseases is based on antiviral regimens to inhibit the virus only. As exemplified by COVID-19, antiviral agent nirmatrelvir has been commonly recommended for treating SARS-CoV-2 but this often fails to effectively cure patients with severe infections[14,15]. Recent evidence suggested that combination of antiviral therapy with the anti-inflammatory corticosteroid drugs may lead to better outcomes in SARS-CoV-2 infected patients[16]. We thus envision that combining antiviral and anti-inflammatory treatments could more effectively manage severe enteric viral infections.

Studying viral pathophysiology and developing advanced treatments require robust experimental models. The development of 3D-cultured intestinal organoids has opened the avenues for better studying intestinal diseases, including infections by enteric viruses (e.g., rotavirus, norovirus)[17–20]. However, these organoids are primarily epithelial structures and inherently lack immune components. Though they can support viral life cycles, they de facto fail to capture the complex virus-host interactions, in particular immune cell-mediated host responses. In fact, the intestinal mucosa is not only lined with epithelial cells, but harbors the largest compartment of immune cells, such as macrophages. Interactions between these immune and epithelial cells are essential for defending against enteric viral infections and maintaining intestinal homeostasis. However, current in vitro models are not competent in recapitulating such interactions between virus and epithelial-immune compartments.

In this study, we established human macrophage-augmented intestinal organoids (MaugOs) that enable to simultaneously recapitulate viral infection and the resultant inflammatory response. We employed echovirus 1, echovirus 6, rotavirus, seasonal coronavirus OC43 and pandemic SARS-CoV-2 as disease modalities. We demonstrated that MaugOs are readily susceptible to the infections and able to model complex virus-host interactions, including the innate antiviral response and the inflammatory response. Taking advantage of MaugOs, we taped into the biological implications and therapeutic potential of gut metabolites in particular acetate, and devised combination treatment encompassing antiviral and anti-inflammatory regimens for treating enteric viral infections.

## Results

### Establishment and characterization of macrophage-augmented intestinal organoids

To establish an in vitro model that can simultaneously simulate enteric viral infections and the resultant inflammatory response, we integrated human macrophages into intestinal-tissue derived organoids, namely 'macrophage-augmented intestinal organoids (MaugOs)' (Fig. 1a). Initially, we used non-activated macrophages differentiated from the THP-1 monocytic cells (Supplementary Fig. 1A). To optimize the cellular viability and biological function of the MaugOs without affecting the migration ability of macrophages, we refined the number of input macrophages, Matrigel concentration, and the recipe of culture medium (Supplementary Fig. 1B-G). Briefly, by utilizing a relatively lower concentration of Matrigel as a supportive base, followed by seeding macrophages and fragmented organoids on the base surface, macrophages and organoids were able to physically migrate and integrate to form MaugOs within 24 hours, as captured by time-lapse video (Fig.1b; Supplementary Movie 1). Using intestinal organoids derived from three different donors, MaugOs models were effectively constructed by optimized protocol (Fig. 1c). The morphology of MaugOs was further visualized by optical imaging, immunofluorescent and immuno-histochemistry (IHC) staining (Fig. 1d; Supplementary Fig. 1H). Stacking Z-dimension images revealed that macrophages integrate into multiple layers of organoids, rather than merely surrounding the surface (Supplementary Fig. 1I; Supplementary Movie 2). To better characterize MaugOs, we performed genome-wide transcriptomic sequencing on THP-1 derived macrophages, organoids and MaugOs (Supplementary Fig. 1J). Analysis of the differentially expressed genes showed that MaugOs express over 9,400 genes common to either macrophages or organoids. Additionally, over ninety genes were exclusively expressed in MaugOs, indicating the active interaction between macrophages and organoids (Fig. 1e; Supplementary Table 1). Importantly, the expression level of key inflammatory genes in MaugOs were comparable to those in separately cultured macrophages and organoids, indicating the macrophages were not activated upon incorporating into organoids (Supplementary Fig. 1K). Pearson correlation analysis further revealed an intermediate gene expression pattern of MaugOs when compared to organoids and macrophages alone (Supplementary Fig. 1L). Intriguingly, KEGG pathway enrichment analysis showed that MaugOs encompassed key physiological profiles representative of organoids and macrophages (Fig. 1f).

To assess the functionality of MaugOs in generating inflammatory response, we treated MaugOs with bacterial lipopolysaccharide (LPS). We first quantified the expression of a panel of inflammatory genes (e.g., IL-1β, IL-6, TNF-α, IL-8). We found that THP-1-derived macrophages, but not organoids, dramatically responded to LPS stimulation (Fig. 1g). Importantly, MaugOs were able to generate robust inflammatory responses to LPS stimuli, with the majority of tested inflammatory genes upregulated from 1 hour to 12 hours post-stimulation (Fig. 1g). This response was further evidenced by significant production of key inflammatory effectors (e.g., IL-1β, IL-6 and IL-8) as quantified by ELISA (Fig. 1h; Supplementary Fig. 1M). To expand the MaugOs model, we established additional MaugOs using macrophages derived from two distinct sources: primary macrophages differentiated from monocytes that were isolated from peripheral blood mononuclear cells (PBMCs) of healthy donors; macrophages differentiated from monocytes that generated from human induced pluripotent stem cells (iPSCs) (Supplementary Fig. 1N-Q). The morphology of both types of MaugOs was visualized by immunofluorescence staining at 36 hours post-seeding (Fig. 1i). Both MaugOs effectively responded to LPS stimulation, with a range of specified inflammatory genes dramatically activated shown by qRT-PCR quantification at 12 hours post-stimulus (Fig. 1j). Consistently, the release of inflammatory effector IL-1β increased by over 10-fold (Fig. 1k). Collectively, functional MaugOs capable of responding to inflammatory stimuli were established.

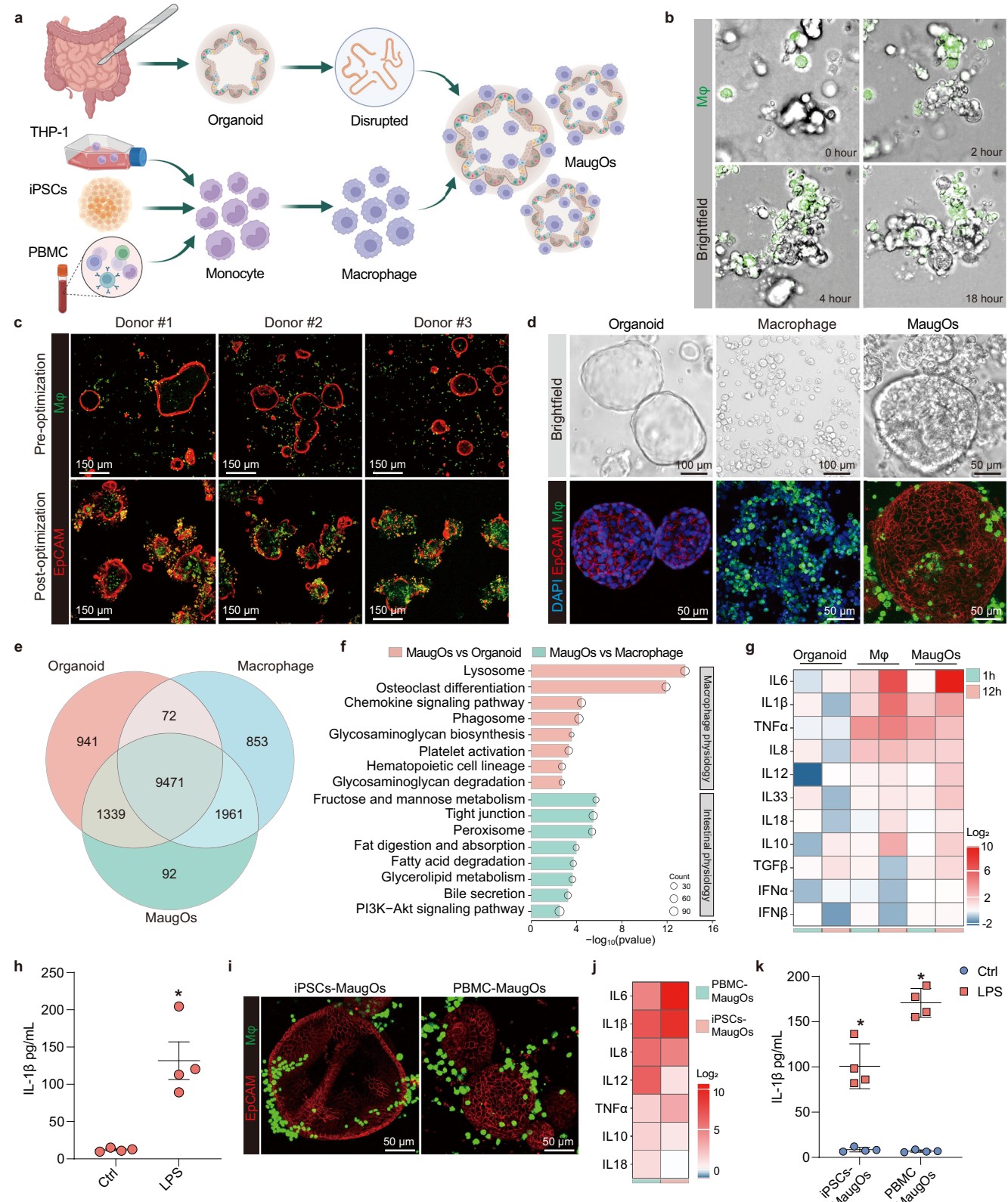

## MaugOs are susceptible to echovirus infections and generate antiviral but not inflammatory response

To evaluate the capability of MaugOs in modelling the pathophysiology of enteric viral infections, we first tested two enteroviruses – echovirus 1 (EV1) and echovirus 6 (EV6), which primarily transmit through the fecal–oral route and infect the intestinal epithelium early during their life cycles[21]. We confirmed that organoids and THP-1

macrophages independently supported the infection of EV1 and EV6, as demonstrated by qRT-PCR quantification of viral RNA and immunostaining of viral replicating dsRNA (Supplementary Fig.2A and 2B). Upon the infection, two echoviruses triggered dramatic antiviral responses in organoids, as evidenced by the expression of tested antiviral interferon-stimulated genes (ISGs) (Supplementary Fig. 2C). However, in THP-1 macrophages, echoviruses failed to elicit

**Fig. 1 | Establishment and characterization of macrophage-augmented intestinal organoids. a** Schematic overview of constructing three different types of macrophage-augmented intestinal organoids (MaugOs). Created in BioRender.com. https://BioRender.com/cegs5ln. **b** Representative images of time-lapse confocal microscopy during MaugOs formation. Unstained intestinal organoid fragments and CFSE pre-labelled macrophages differentiated from THP-1 monocytes were integrated for 18 hours. Images were taken every 30 minutes (also see Supplementary Movie 1). **c** Immunofluorescence staining images of MaugOs incorporating organoids from three donors. Epithelial membrane of organoids was stained by EpCAM (red); THP-1 monocytes-derived macrophages stained by CFSE (green). Scale bar, 150 μm. **d** Bright field and immunofluorescence staining images depicting organoids, THP-1 macrophages and MaugOs. **e** Venn diagram of overlapped differentially expressed genes in macrophages (Mφ), intestinal organoids and MaugOs. **f** Significantly enriched pathways by KEGG analysis of MaugOs. ($p < 0.05$). **g** The expression of inflammatory genes quantified by qRT-PCR upon

stimulation of 1 μg/mL LPS for 1 hour and 12 hours. Each group was compared to its respective control without LPS treatment ($n = 3$). **h** Quantification of IL-1β protein production by ELISA in MaugOs integrated THP-1 monocytes-derived macrophages ($n = 4$). **i** Immunofluorescence staining images of MaugOs integrated PBMC monocytes-derived macrophages and iPSCs-derived macrophages. Epithelial membrane of organoids was stained by EpCAM (red); PBMC monocytes-derived macrophages were stained by CFSE, and iPSCs-derived macrophages expressed H2B-mEGFP protein (green). Scale bar, 50 μm. **j** The expression of inflammatory genes quantified by qRT-PCR upon stimulation of LPS in MaugOs integrated iPSCs-derived or PBMC monocytes-derived macrophages ($n = 2$). **k** Quantification of IL-1β protein production by ELISA in MaugOs integrated PBMC monocytes-derived macrophages and iPSCs-derived macrophages ($n = 4$). Data were presented as means of biological replicates ± SEM. Statistical analysis was performed using the two-tailed Mann–Whitney test. *$p < 0.05$.

inflammatory responses despite of robust viral replication (Supplementary Fig. 2D). We then assembled MaugOs by integrating echovirus pre-inoculated organoids with THP-1 macrophages. QRT-PCR quantifying viral RNA revealed over thousands-fold increase of EV1 and EV6 replication in MaugOs from 1 to 36 hours post-MaugOs assembly (Figs. 2a and b). TCID$_{50}$ assay detected an over 100-fold increase of viral infectious titers in the culture supernatant for both EV1 and EV6 (Figs. 2c and d). Immunostaining viral replicating dsRNA further confirmed the robust infection (Figs. 2e and f). Importantly, both EV1 and EV6 infections in MaugOs elicited antiviral responses, with ISGs dramatically upregulated in a time-dependent manner (Fig. 2g). Despite the robust infection, there was minimal activation of inflammatory responses in MaugOs, as shown by qRT-PCR quantifying the expression of key inflammation-associated genes (Fig. 2g). Overall, MaugOs showed high susceptibility to echovirus infections, which induced pronounced antiviral but limited inflammatory responses.

## Rotavirus and coronavirus OC43 infections in MaugOs trigger both antiviral and inflammatory responses

To further evaluate the ability of MaugOs to model distinct virus-host interactions, we employed rotavirus as the modality. We first observed that both organoids and THP-1 macrophages independently support rotavirus infections (Supplementary Fig. 2E and F). This infection triggered antiviral responses in organoids and inflammatory responses in THP-1 macrophages (Supplementary Fig. 2G). Next, rotavirus pre-inoculated organoids were integrated with THP-1 macrophages and assembled into MaugOs. Quantification of viral RNA by qRT-PCR revealed consistent replication of rotavirus in MaugOs from 1 to 36 hours post-assembly (Fig. 2h). This robust replication resulted in significant production of infectious viruses in the supernatant as quantified by TCID$_{50}$ assay (Fig. 2i). Intriguingly, immunostaining rotavirus VP6 protein showed virus transmission from initially infected organoids to neighboring macrophages (Fig. 2j). Notably, this infection triggered the dramatic expression of antiviral ISGs from 12 to 36 hours post-inoculation (Fig. 2k). Moreover, rotavirus infection elicited inflammatory responses, as shown by the significant *IL-1β* gene expression and production (Fig. 2l and Supplementary Fig. 2H). Subsequently, we tested coronavirus OC43 as it has been shown to cause intestinal manifestations in infected patients[6]. We found that organoids, THP-1 macrophages and MaugOs all support OC43 infections and infectious virus production (Supplementary Fig. 2I and 2J; Fig. 2m-o). Similar to rotavirus, OC43 virus infection as well triggered antiviral responses in organoids and inflammatory responses in THP-1 macrophages (Supplementary Fig. 2K). In MaugOs, OC43 virus was capable of inducing potent antiviral and inflammatory responses at 36 hours post-inoculation (Figs. 2p and q; Supplementary Fig. 2L). In particular the inflammatory response, we observed over 10-fold higher IL-1β elevation in both RNA and protein level triggered by OC43 virus compared to that by rotavirus (Fig. 2q; Fig. L). Quantification of the IL-6, IL-8 and TNF-α

production further confirmed the robust inflammatory response by OC43 infection (Supplementary Fig. 2M). Additionally, we tested OC43 virus infection in MaugOs integrating either PBMCs- or iPSCs-derived macrophages. Similarly, both types of MaugOs supported OC43 virus infections, and the infection triggered significant IL-1β production (Supplementary Fig. 2N-Q). Lastly, we tested OC43 virus infection in MaugOs incorporating organoids from two additional donors. As expected, these MaugOs models were readily supportive to OC43 infection (Fig. 2r). Interestingly, intracellular viral copy numbers appeared slightly higher in MaugOs-donor#3 than in MaugOs-donor#2 at 36 hours post-infection, although this was not statistically significant (Fig.2r). Both MaugOs exhibited robust antiviral and inflammatory responses, with MaugOs-donor#3 displaying higher expression of certain inflammation- and antiviral-associated genes, such as *IL10*, *IFIT1* and *STAT1*, compared to MaugOs-donor#2 (Fig. 2S). These results demonstrated that MaugOs were suitable for studying virus-host interactions.

## Characterizing the complex virus-host interactions in MaugOs

To further characterize virus-host interactions, we selected coronavirus OC43 and EV1 as disease modalities and performed the genome-wide transcriptomic analysis for infected and non-infected THP-1 macrophages, organoids, and MaugOs-integrated THP-1 macrophages. RNA sequencing mapped the reads of virus genomic fragments across the EV1 and OC43 viral genomes, revealing the potent viral infections in these models (Fig. 3a; Supplementary Fig. 3A). Principal component analysis (PCA) distinctly separated the infected from non-infected organoids and MaugOs. However, EV1-infected THP-1 macrophages largely overlapped with non-infected macrophages (Supplementary Fig. 3B), indicating the minimal influence of EV1 infection to macrophages. Intriguingly, interactive Venn diagrams revealed over one hundred genes uniquely expressed in MaugOs compared to organoids and macrophages upon infection with either OC43 or EV1 (Supplementary Fig. 3C; Supplementary Table 2 and 3). Likewise, thousands of genes were specifically regulated in EV1- or OC43-infected MaugOs compared to uninfected MaugOs (Supplementary Fig. 3D). These results indicated the active interplay between macrophages and organoids in MaugOs following viral infections. Volcano plots revealed that the most highly upregulated genes in response to OC43 infection, such as *SOD2*, *CD44*, *STAT1* and *MX1*, were inflammation-associated genes and antiviral ISGs (Fig. 3b). KEGG analysis identified significant upregulation of multiple antiviral and inflammatory-associated pathways including NF-κB signaling pathway and NOD-like receptor signaling pathway upon OC43 infection (Supplementary Fig. 3E). In contrast, EV1 predominantly activated antiviral ISGs and downregulated intestinal physiology associated genes (Fig. 3c). Gene set enrichment analysis revealed enrichment of apoptosis-associated transcriptional signature in MaugOs upon EV1 and OC43 infections (Fig. 3d). We then performed propidium iodide (PI) staining in MaugOs following EV1 and OC43 infections. Importantly, many

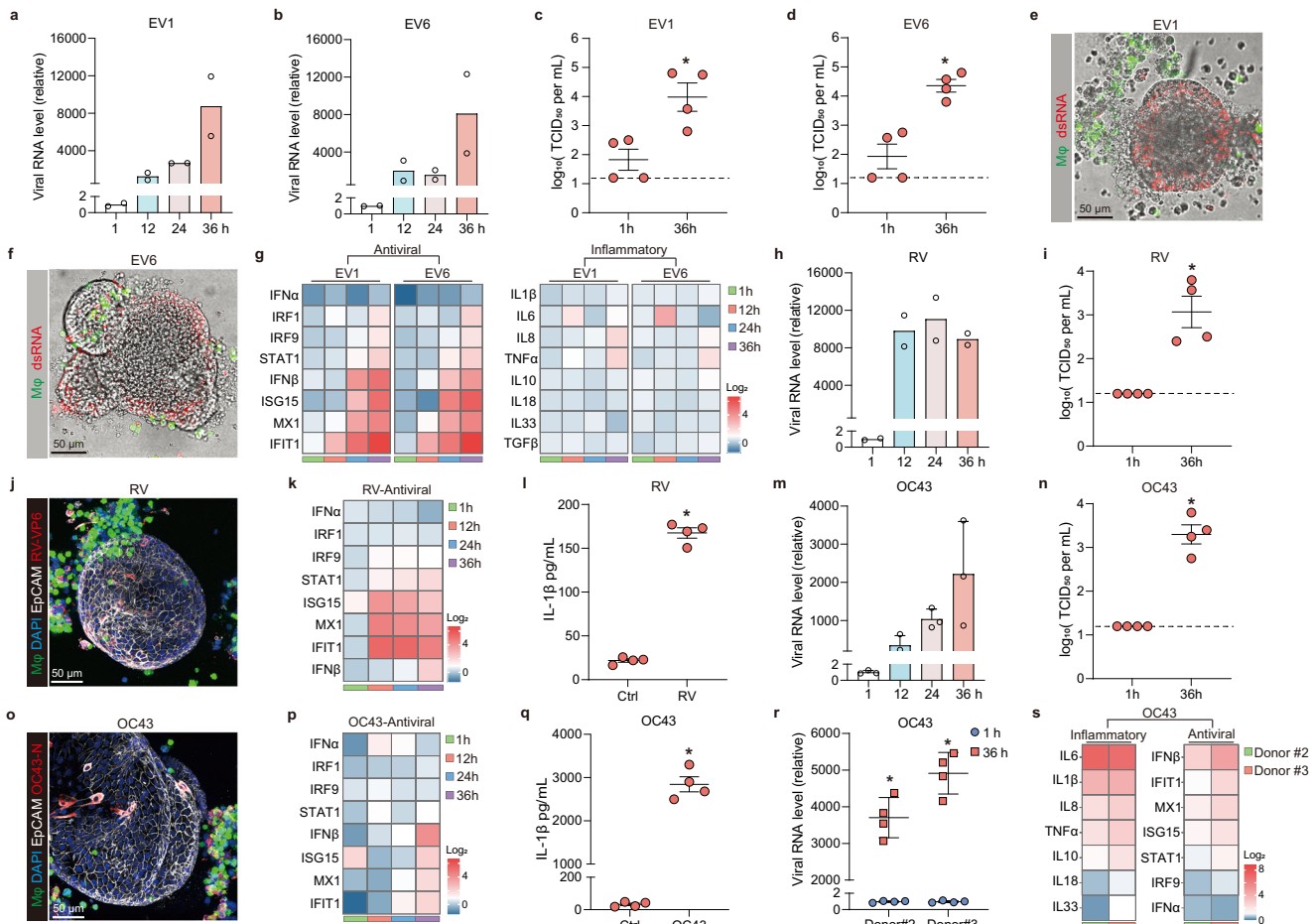

**Fig. 2 | MaugOs recapitulate enteric viral infections and virus-host interactions.**
**a**–**b** Viral RNA level in EV1 (**a**) or EV6 (**b**) infected MaugOs from 1 to 36 hours post-infection (hpi). Data was normalized to viral RNA level at 1 hpi ($n = 2$). **c**–**d** Quantification of infectious EV1 (**c**) and EV6 (**d**) titers in culture supernatants from 1 to 36 hpi ($n = 4$). Dotted line indicates detection limit. **e**–**f** Confocal imaging of viral replicating dsRNA (red) in EV1 (**e**) or EV6 (**f**) infected MaugOs at 36 hpi. Scale bar, 50 μm. **g** RNA expression of ISGs and inflammatory genes in EV1- or EV6-infected MaugOs from 1 to 36 hpi ($n = 2$). **h** Viral RNA level in rotavirus-infected MaugOs from 1 to 36 hpi ($n = 2$). Data was normalized to viral RNA level at 1 hpi.
**i** Quantification of infectious rotavirus titers in culture supernatants from 1 to 36 hpi ($n = 4$). **j** Confocal imaging of rotavirus VP6-protein (red) in MaugOs at 36 hpi. Scale bar, 50 μm. **k** ISGs expression in rotavirus-infected MaugOs from 1 to 36 hpi ($n = 2$). **l** Quantification of IL-1β that secreted from uninfected- and rotavirus-infected

MaugOs ($n = 4$). **m** Viral RNA expression in OC43-infected MaugOs from 1 to 36 hpi ($n = 3$). Data was normalized to viral RNA level at 1 hpi. **n** Quantification of infectious OC43 titres in culture supernatants from 1 to 36 hpi ($n = 4$). **o** Confocal imaging of OC43 N-protein (red) in MaugOs at 36 hpi. Scale bar, 50 μm. **p** ISGs expression in OC43-infected MaugOs from 1 to 36 hpi ($n = 3$). **q** Quantification of IL-1β that secreted from uninfected- and OC43-infected MaugOs ($n = 4$). **r** Viral RNA expression in OC43-infected MaugOs incorporating organoids from two additional donors from 1 to 36 hpi ($n = 4$). Data was normalized to RNA level at 1 hpi. **s** ISGs and inflammatory genes expression in OC43-infected MaugOs incorporating organoids from two additional donors at 36 hpi ($n = 2$). All data were presented as means of biological replicates ± SEM. Statistical analysis was performed using the two-tailed Mann–Whitney test. *$p < 0.05$.

PI-positive cells were identified in EV1 and OC43 infected MaugOs, indicating the cell death induced by virus infections (Fig. 3e). We further measured lactate dehydrogenase (LDH), a classical indicator of cellular damage. Consistently, both EV1 and OC43 triggered significant LDH release in MaugOs (Fig. 3f). Collectively, these results suggested that MaugOs were capable of generating distinct levels of host responses and studying complex virus-host interactions.

Next, we explored the NLRP3 inflammasome activation in MaugOs with OC43 virus as modality, since OC43 virus triggered the most robust inflammatory responses among tested viruses. Two signals are typically required to trigger NLRP3 inflammasome cascade. Initially, NF-κB pathway is activated and mediates the transcription of NLRP3 and pro-IL-1β. Secondly, primed NLRP3 inflammasome induces the auto-activation of caspase-1, resulting in the cleavage of pro-IL-1β and the secretion of mature IL-1β[22]. Consistently, we observed elevated levels of NF-κB protein upon OC43 inoculation in MaugOs integrated THP-1 macrophages (Fig. 3g). Western blotting demonstrated the time-dependently increased level of NLRP3, pro IL-1β and

pro caspase-1 intracellularly, and the secretion of cleaved IL-1β and cleaved caspase-1 in supernatant (Fig. 3g). In addition, the phosphorylated signal transducer and activator of transcription 1 (p-STAT1), interferon regulatory factor 9 (IRF-9), protein kinase R (PKR), and phosphorylated eukaryotic initiation factor 2 alpha (p-eIF2α) are pivotal components of the host antiviral responses[23]. Likewise, these antiviral elements were profoundly activated by OC43 and peaked at 36 hours post-infection in MaugOs (Fig. 3g). In addition, we established a trans-well culture system, with THP-1 monocytes differentiated macrophages seeding in the apical insert and OC43 virus-infected organoids growing in the basolateral compartment (Fig. 3h). Immunostaining the OC43-N protein at 48 hours post-infection revealed potent virus infection in organoids but also macrophages in the apical insert, indicating a viral transmission from organoids to macrophages through the paracrine route (Supplementary Fig. 3F). This secondary infection in macrophages also activated profound inflammatory response (Fig. 3i; Supplementary Fig. 3G). Intriguingly, this system enables macrophages to migrate to the backside of trans-well

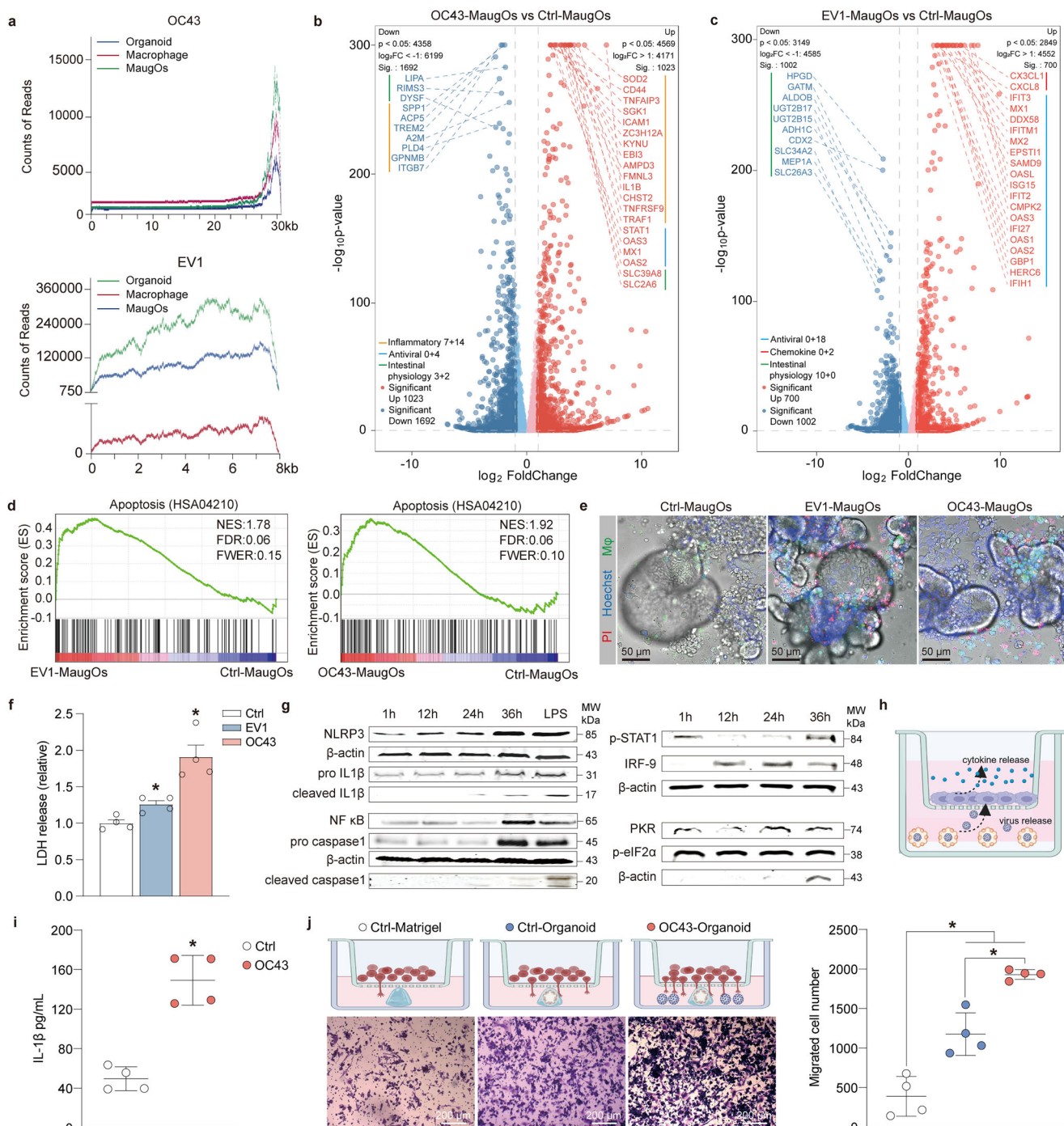

**Fig. 3 | Characterizing the complex virus-host interactions in MaugOs. a** Whole genome view of mapped transcripts by RNA-seq across the OC43 or EV1 genome in infected organoid, macrophage and MaugOs. **b–c** Differential gene expression analysis of OC43 infection (**b**) and EV1 infection (**c**) in MaugOs compared to uninfected MaugOs ($n = 3$). **d** Gene set enrichment analysis (GSEA) of KEGG apoptosis pathways (HSA04210) in MaugOs infected with OC43 or EV1. **e** Fluorescence staining of dead cells (PI, red), cell nuclei (Hoechst, blue), and bright field images at 36 hours post-inoculation following OC43 or EV1 infection in MaugOs. CFSE-labeled THP-1-derived macrophages are shown in green. Scale bar, 50 μm. **f** Quantification of lactate dehydrogenase (LDH) release in MaugOs upon OC43 or EV1 infection at 36 hours post-inoculation ($n = 4$). MaugOs without virus infection served as control (normalized as 1). **g** OC43-infected organoids were integrated with THP1 macrophages and samples were collected at 1 hour, 12 hour, 24 hour and 36 hour after MaugOs assembly. The protein level of NLRP3, pro IL-1β, NF-κB and pro caspase 1 in

lysates, and cleaved IL-1β mature and cleaved caspase-1 in supernatant were determined by western blotting. LPS treatment for 36 hours in MaugOs was used as positive control. The protein level of p-STAT1, IRF-9, p-eIF2α in MaugOs lysates were determined by western blotting at 1 hour, 12 hour, 24 hour, and 36 hour post-OC43 infection. **h** Schematic representation of OC43-infected organoids culturing in the down compartment and THP-1 macrophages culturing in the transwell insert. Created in BioRender.com. https://BioRender.com/3vjrdll. **i** ELISA quantification of IL-1β released into the insert supernatant by macrophages ($n = 4$). **j** Cell migration assay determining the macrophage number attracted by Matrigel (as control), uninfected and OC43-infected organoids cultured in Matrigel ($n = 4$). Scale bar, 200 μm. Created in BioRender.com. https://BioRender.com/jv8nl58. All data were presented as means of biological replicates ± SEM. Statistical analysis was performed using the two-tailed Mann–Whitney test. *$p < 0.05$.

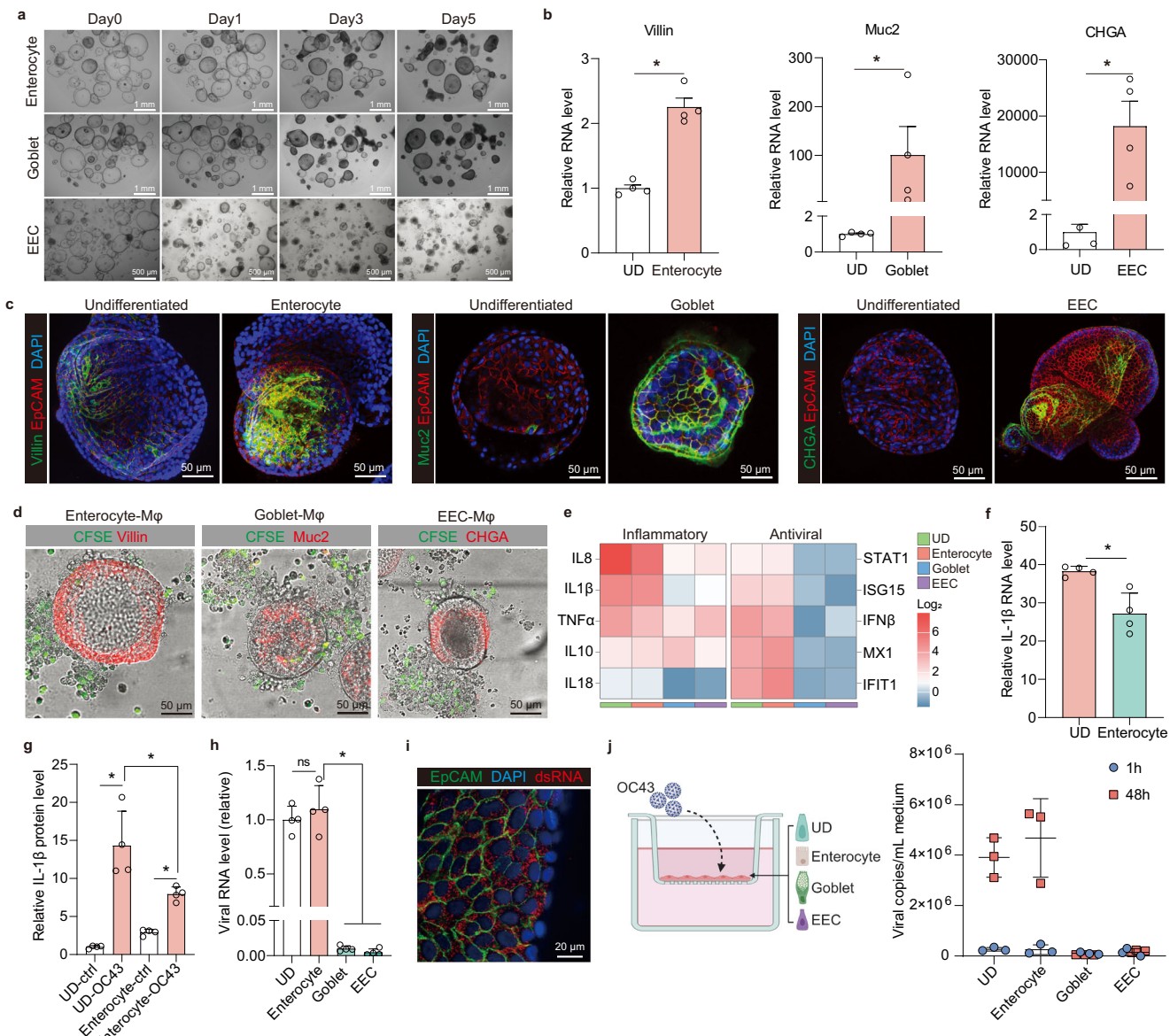

**Fig. 4 | Establishment of MaugOs using differentiated intestinal organoids.**
**a** Morphology of organoids differentiating towards enterocytes-, goblet- and enteroendocrine (EEC)-phenotypes in different time points. **b** Gene expression level of *Villin* (enterocytes marker), *Muc2* (goblet cell marker) and *CHGA* (enteroendocrine cell marker) upon different differentiation culture (*n* = 4). UD: undifferentiated organoids; EEC: enteroendocrine-differentiated organoids.
**c** Characterizing cell types of differentiated organoids by immunofluorescence staining of Villin (enterocytes), Muc2 (goblet cells) and CHGA (enteroendocrine cells). EpCAM is the marker of epithelial cell membrane (red). Scale bar, 35 μm and 50 μm. **d** Morphology of MaugOs integrated with enterocytes- (Villin, red), goblet cells- (Muc2, red) and enteroendocrine cells (EEC)- (CHGA, red) differentiated organoids. THP1 macrophages were stained with CFSE (green). Scale bar, 50 μm.
**e** QRT-PCR quantification of inflammatory and antiviral associated genes

expression in MaugOs integrated differentiated or undifferentiated organoids upon OC43 infection (*n* = 2). **f** Quantification of *IL-1β* gene expression in MaugOs integrated undifferentiated or enterocytes-differentiated organoids (*n* = 4). **g** ELISA quantification of IL-1β protein production in MaugOs integrated undifferentiated or enterocytes-differentiated organoids (*n* = 4). **h** Quantification of OC43 viral RNA level in MaugOs integrated undifferentiated or differentiated organoids at 36 hours postinfection (*n* = 4). **i** Confocal imaging of viral replicating double-stranded RNA (dsRNA, red color) in enterocytes-differentiated organoids, at 48 hours post OC43 infection. Scale bar, 20 μm. **j** QRT-PCR quantification of released OC43 virus from polarized intestinal monolayers cultured in transwell system (*n* = 3). Created in BioRender.com. https://BioRender.com/yz1fjns. All data were presented as means of biological replicates ± SEM. Statistical analysis was performed using the two-tailed Mann–Whitney test. \**p* < 0.05.

membrane. We observed significantly elevated numbers of macrophages on the backside when organoids were infected with OC43, compared to Matrigel alone or non-infected organoids (Fig. 3j).

### MaugOs based on enterocytes-differentiated organoids support OC43 virus infection, and generate antiviral and inflammatory responses

Intestinal organoids are cultured from stem cells residing at the base of intestinal crypts. These stem cells can further differentiate into

different epithelial cell types[24]. To further establish MaugOs encompassing different epithelial cell types, organoids were undergone differentiation culture towards enterocytes, goblet cells and enteroendocrine cells. Upon 5 days of differentiation, organoids showed a darker appearance with thicker walls (Fig. 4a). Gene expression of specific cell markers including *Villin, Mucin 2* (*MUC2*) and *Chromogranin-A* (*CHGA*) were significantly upregulated in three types of differentiated organoids (Fig. 4b). Consistently, immunofluorescence staining showed more enterocytes (Villin-positive),

goblet cells (Muc2-positive), and enteroendocrine cells (CHGA-positive) in three differentiated organoids (Fig. 4c). Next, we integrated THP-1 derived macrophages with differentiated organoids for assembling MaugOs, as visualized by immunofluorescence microscopy (Fig. 4d; Supplementary Fig. 4A). To further expand the application of these models for studying viral infections, we inoculated MaugOs integrating differentiated organoids with OC43 as a representative virus. We observed that OC43 infection activated robust expression of inflammatory genes in MaugOs integrated undifferentiated or enterocytes-differentiated organoids, and mild to moderate activation of inflammatory genes expression in MaugOs integrated goblet- or enteroendocrine-differentiated organoids. In parallel, a panel of ISGs was activated by OC43 infection in MaugOs integrated with undifferentiated or enterocytes-differentiated organoids, whereas MaugOs integrated with goblet- or enteroendocrine- differentiated organoids hardly presented antiviral response to the infection (Fig. 4e). The robust inflammatory response in MaugOs integrated with enterocyte-differentiated organoids was further confirmed by significantly elevated IL-1β production, albeit comparatively lower than that in MaugOs integrated with undifferentiated organoids (Figs. 4f and g).

We speculated that the differentiated organoids might encompass distinct susceptibility to OC43 virus, which resulted in discrepant antiviral and inflammatory response in MaugOs. We thus compared the infection effect between different MaugOs and found that OC43 viral RNA level was significantly lower in MaugOs integrated with goblet- or enteroendocrine-differentiated organoids compared to that in undifferentiated organoids (Fig. 4h). Similarly, the solely infection to organoids demonstrated that OC43 virus can efficiently replicate in enterocytes-differentiated organoids, but not goblet- or enteroendocrine-differentiated organoids (Fig. 4i; Supplementary Fig. 4B and C). To further assess the infectivity of OC43 virus, organoids were transformed into monolayer by seeding in transwell inserts and further differentiated to different cell types. The transepithelial electrical resistance (TEER) of monolayers was peaked at ten days and displayed intact integrity by staining epithelial membrane marker EpCAM (Supplementary Fig. 4D and E). The stabilized and intact monolayers were then inoculated with OC43 virus from apical compartments as shown by the schematic diagram (Fig. 4j). We observed a dramatic increase of viral copies from 1 hour to 48 hours post-inoculation in undifferentiated and enterocytes-differentiated monolayers, and the infected monolayers secreted predominate viruses into the apical but not basolateral compartment (Fig. 4j; Supplementary Fig. 4F). In contrast, differentiated goblet and enteroendocrine cells were not supportive to OC43. Collectively, MaugOs integrating enterocytes-differentiated organoids, but not goblet- or enteroendocrine-differentiated organoids, were readily supportive to OC43 infection and study virus-host interactions.

## Mapping the pleiotropic effects of acetate on viral infection and inflammatory response in MaugOs

Short chain fatty acids (SCFAs), mainly consisting of acetate, butyrate and propionate, are a typical group of microbial metabolites that have marked impact on intestinal homeostasis. To unravel the role of SCFAs on the inflammatory responses, we tested acetate, propionate and butyrate in bacterial LPS elicited-MaugOs. We first employed LPS-elicited-MaugOs consisting of intestinal organoids and THP-1 macrophages. By quantifying the inhibitory effect to specified inflammatory genes expression at 12 hours post-treatment, we found that compared to propionate and butyrate, acetate exerted the most robust anti-inflammatory activity, which dramatically blocked the upregulation of all tested inflammatory genes without cytotoxicity (Fig. 5a; Supplementary Fig. 5A). This robust anti-inflammatory effect was also observed at 24 hours post-treatment (Supplementary Fig. 5B). Likewise, the production of pro-inflammatory cytokines triggered by LPS, including IL-1β, IL-6 and IL-8, was significantly blocked by acetate

treatment (Fig. 5b–d). Afterwards, we assessed the anti-inflammatory activity of acetate in MaugOs challenged with OC43 virus. Similarly, the expression of OC43-elicited inflammatory genes was significantly abrogated in RNA level (Supplementary Fig. 5C), and the production of corresponding inflammatory effectors was significantly inhibited (Fig. 5e–h). The potent inhibition to inflammatory response was further observed when challenging with rotavirus (Fig. 5i; Supplementary Fig. 5D). Next, in MaugOs, respectively, integrated PBMC- and iPSCs-derived macrophages, comparable anti-inflammatory effect of acetate was observed when stimulating the model with LPS treatment or OC43 inoculation (Fig. 5j, k). Clinically, secondary bacterial infections often emerge following acute viral infections, thereby concurrently contributing to the pathological inflammation in patients. To further recapitulate this scenario, OC43-elicited MaugOs were simultaneously treated with bacterial LPS. Notably, this resulted in a synergistic effect on the level of activated inflammatory response, with approximately three-fold higher IL-1β expression and production compared to that triggered by LPS or OC43 alone (Fig. 5l). Importantly, acetate remains its robust inhibitory activity against this augmented inflammatory response (Fig. 5l).

To specifically dissect the effect of acetate on viral infection in different component of MaugOs, we separately analyzed OC43 infection in organoids, TPH-1 macrophages, and MaugOs. Interestingly, the viral replication was significantly promoted in organoids but inhibited in THP-1 macrophages upon 36 hours treatment of acetate (Fig. 5m, n). This disparity resulted in a minimal-to-mild inhibitory effect of acetate to viral replication in MaugOs (Fig. 5o; Supplementary Fig. 5E). Similar to OC43 infection in organoids, acetate exerted a promoting effect to the replication of rotavirus, EV1 and EV6 in organoids (Fig. 5p, q). In addition, we investigated the influence of acetate to the cell polarization in organoids. Upon 36 hours treatment of acetate in organoids, the expression of enterocytes marker (Villin-positive) was significantly upregulated and goblet cell marker (Muc2-positive) was downregulated (Supplementary Fig. 5F), which may partially explain the enhanced viral replication in organoids. Interestingly, both acetate treatment and OC43 virus infection were able to upregulate the RNA level of enterocyte marker Villin in MaugOs, with their combination further synergizing the expression level (Supplementary Fig. 5G). These results indicated that acetate and OC43 virus may facilitate enterocyte differentiation in MaugOs. We next performed transcriptional analysis for mapping the influence of acetate in MaugOs (Supplementary Fig. 5H). Differential gene expression analysis showed that the impact of rewiring host transcriptome by OC43 infection, predominantly including antiviral and inflammatory-associated genes, was largely prevented by acetate treatment (Fig. 5r). Interestingly, KEGG and differential gene expression analysis revealed that acetate upregulated anti-inflammatory M2 macrophage-associated genes and pathways, and downregulated pro-inflammatory M1 macrophage-associated genes and pathways (Fig. 5s, Supplementary Fig. 5I), indicating the transition of macrophages from M1 to M2 in MaugOs upon acetate treatment. To broaden the implication of NLRP3 inflammasome as a therapeutic target, we employed MaugOs activated by bacterial LPS or OC43 virus. First, we confirmed the activation of the NLRP3 inflammasome cascade in MaugOs. Western blotting indicated increased levels of NF-κB, NLRP3, pro IL-1β and pro caspase-1 intracellularly, along with cleaved IL-1β and cleaved caspase-1 in the supernatant (Fig. 5t, u). Of note, this activation can be specifically blocked by treatment with BAY11-7085 (NF-κB inhibitor), MCC950 (NLRP3 inhibitor) or VX-765 (caspase-1 inhibitor). These pharmaceutical inhibitors also significantly reduced the production of inflammatory effectors (e.g., IL-1β, IL6, IL8) (Supplementary Fig. 5J-M). Next, we assessed the role of acetate in the NLRP3 inflammasome cascade. Importantly, we found that acetate presented a comparable effect to the aforementioned pharmaceutical inhibitors, showing dramatic inhibition to NF-κB, NLRP3, pro IL-1β, pro caspase-1, cleaved IL-1β and

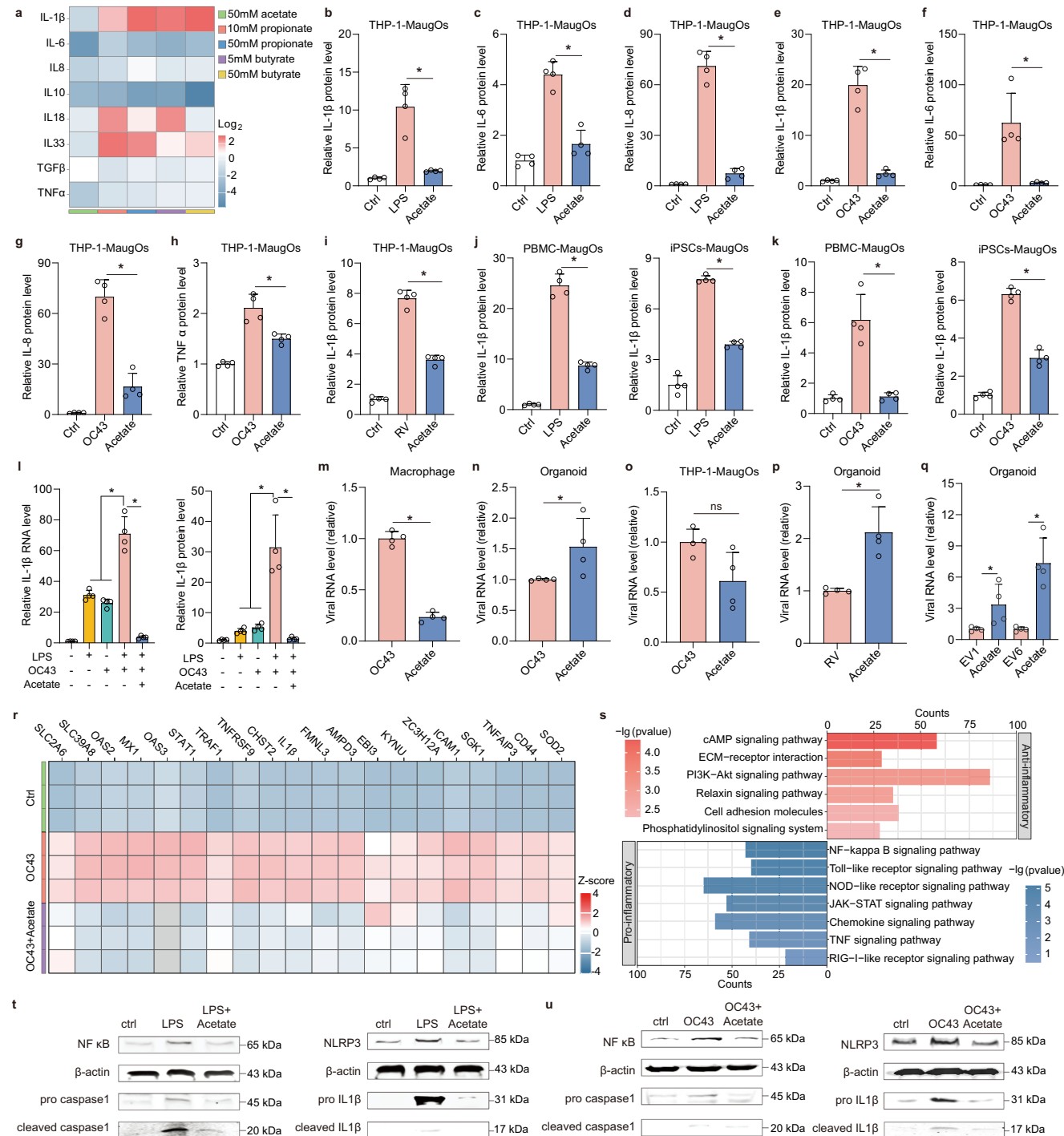

**Fig. 5 | Characterizing the function of acetate on enteric viral infection and inflammatory response in MaugOs. a** Profiling the influence of three metabolites on inflammatory gene expression triggered by LPS in MaugOs ($n = 2$). **b–d** ELISA quantification of IL-1β (**b**), IL-6 (**c**) and IL-8 (**d**) in the supernatant of MaugOs following LPS stimulation with or without 50 mM acetate treatment for 24 hours ($n = 4$). **e–h** ELISA quantification of IL-1β (**e**), IL-6 (**f**), IL-8 (**g**) and TNF-α (**h**) in the supernatant of MaugOs following OC43 infection with or without 50 mM acetate treatment for 36 hours ($n = 4$). **i** Quantification of IL-1β production in the supernatant of rotavirus-infected MaugOs with or without 50 mM acetate treatment for 36 hours ($n = 4$). **j–k** Quantification of IL-1β production in the supernatant of MaugOs integrated PBMC- or iPSCs-derived macrophages after LPS stimulation (**j**) or OC43 infection (**k**), with or without 50 mM acetate treatment for 24 hours ($n = 4$). **l** The inhibitory effect of 50 mM acetate treatment for 12 hours on the IL-1β gene

expression and protein production in MaugOs simultaneously elicited by LPS and OC43 infection ($n = 4$). **m–o** Quantification of viral RNA level in OC43-infected THP1 macrophages (**m**), organoids (**n**) and MaugOs (**o**) with or without 50 mM acetate treatment for 36 hours ($n = 4$). **p–q** Quantification of viral RNA level in rotavirus-infected (**p**), EV1 or EV6-infected (**q**) organoids with or without 50 mM acetate treatment for 36 hours ($n = 4$). **r** Top 20 significantly regulated genes upon OC43 infection in MaugOs ($n = 3$). **s** Significantly regulated pathways by KEGG analysis at 36 hours in OC43-infected MaugOs with 50 mM acetate treatment ($p < 0.05$), compared with the non-treatment group. Red: upregulated; blue: downregulated ($n = 3$). **t–u** The inhibitory effect of 50 mM acetate treatment on the protein level of NF-κB and NLRP3 signaling cascade in LPS-treated (**t**) or OC43-infected (**u**) MaugOs. All data were presented as means of biological replicates ± SEM. Statistical analysis was performed using the two-tailed Mann–Whitney test. *$p < 0.05$.

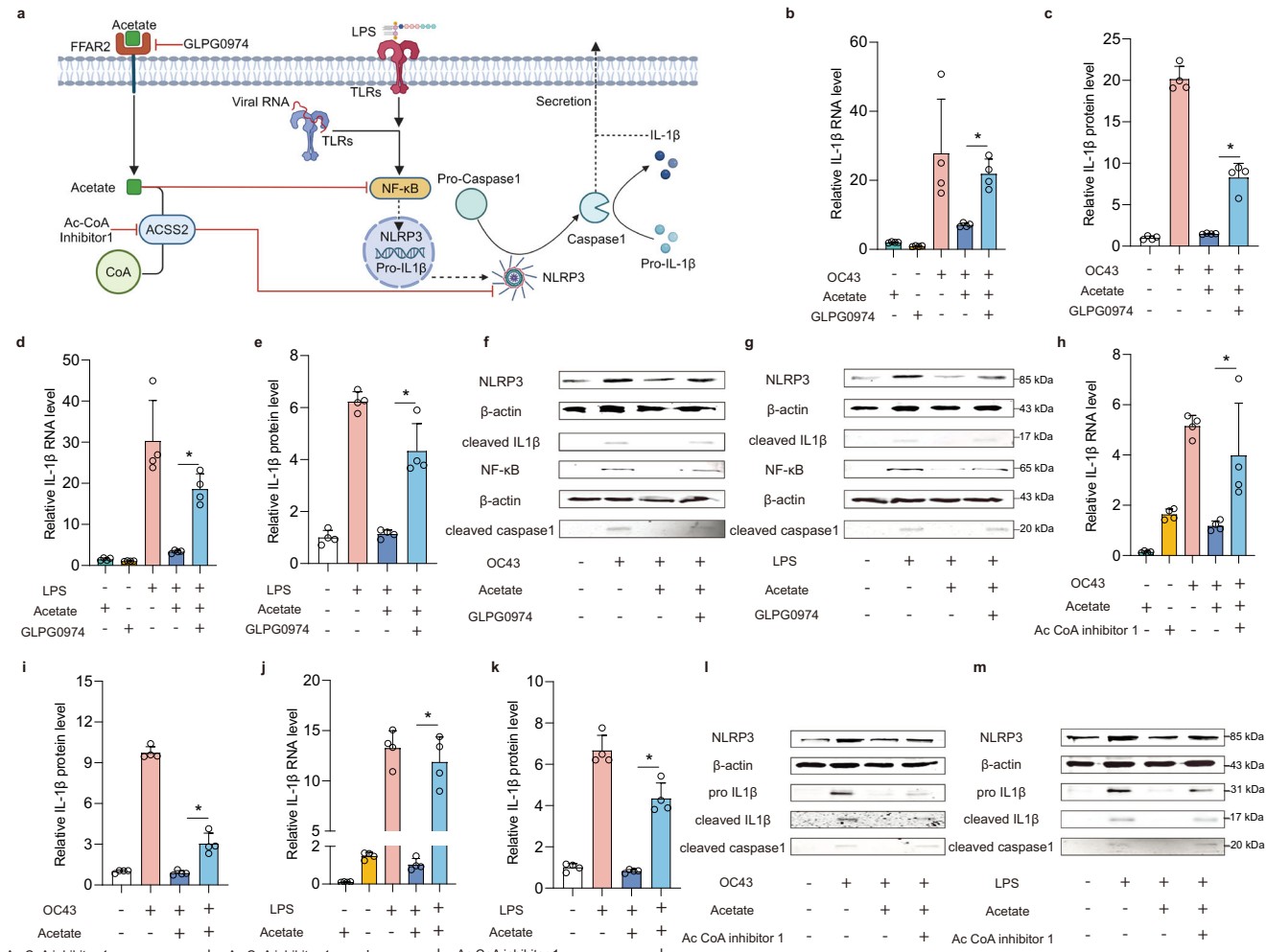

**Fig. 6 | Dissecting the anti-inflammatory effects of acetate in enteric infections: mechanism-of-action. a** Schematic illustration of the key receptor and signaling pathways that targeted by acetate. Created in BioRender.com. https://BioRender.com/0vfnx9c. **b** The effect of GLPG0974 on the expression of *IL-1β* in OC43 virus-infected MaugOs quantified by qRT-PCR (*n* = 4). **c** The effect of GLPG0974 on the production of IL-1β in the supernatant of OC43-infected MaugOs quantified by ELISA (*n* = 4). **d** The effect of GLPG0974 on the expression of *IL-1β* in LPS-treated MaugOs quantified by qRT-PCR (*n* = 4). **e** The effect of GLPG0974 on the production of IL-1β in the supernatant of LPS-treated MaugOs quantified by ELISA (*n* = 4). **f** and **g** The effect of GLPG0974 on the protein level of NLRP3 signaling cascade in OC43-infected (**f**) or LPS-treated (**g**) MaugOs at 36 hours determined by western blotting. NLRP3 and NF-κB were detected from cell lysates; Cleaved caspase-1 and cleaved IL-1β were detected from culture supernatant. **h** The effect of AC-CoA inhibitor 1 on the expression of *IL-1β* in OC43-infected MaugOs quantified by qRT-PCR (*n* = 4). **i** The effect of AC-CoA inhibitor 1 on the production of IL-1β in the supernatant of OC43-infected MaugOs quantified by ELISA (*n* = 4). **j** The effect of AC-CoA inhibitor 1 on the expression of *IL-1β* in LPS-treated MaugOs quantified by qRT-PCR (*n* = 4). **k** The effect of AC-CoA inhibitor 1 on the production of IL-1β in the supernatant of LPS-treated MaugOs quantified by ELISA (*n* = 4). **l** and **m** The effect of AC-CoA inhibitor 1 on the protein level of NLRP3, pro IL-1β and cleaved caspase 1 in OC43-infected (**l**) or LPS-treated (**m**) MaugOs at 36 hours determined by western blotting. All data were presented as means of biological replicates ± SEM. Statistical analysis was performed using the two-tailed Mann–Whitney test. **p* < 0.05.

cleaved caspase-1 (Fig. 5t, u). These collectively demonstrated the potential of acetate as a robust anti-inflammatory agent targeting the NLRP3 inflammasome.

## Dissecting the mechanism-of-action of acetate-mediated anti-inflammatory activity

Mechanistically, acetate signals from the extracellular compartment to the cytoplasm through directly activating the free fatty acid receptor-2 (FFAR2). Once in the cytoplasm, acetate interacts with Acyl-coenzyme A synthetase short-chain family member 2 (ACSS2) to form Acetyl-CoA[25], which performs beneficial functions such as preventing NLRP3 inflammasome activation (Fig. 6a). To probe the anti-inflammatory mechanism of acetate, LPS- or OC43 virus- elicited MaugOs were treated with GLPG0974—a FFAR2 antagonist. Notably, the inhibition of LPS or OC43-triggered inflammatory response by acetate was largely reversed by GLPG0974, showing significantly elevated IL-1β RNA expression and protein production compared to acetate treatment alone (Fig. 6b–e). Consistently, acetate-induced inhibition of NLRP3 inflammasome cascade, including NLRP3, NF-κB, cleaved IL-1β and cleaved caspase 1, was dramatically abrogated by GLPG0974 treatment (Fig. 6f, g). Next, we pharmaceutically blocked ACSS2 by treating with Ac-CoA Synthase Inhibitor1. In LPS- or OC43 virus-activated MaugOs, the blockade of ACSS2 dramatically abrogated the inhibitory effect of acetate on the expression and production of IL-1β (Fig. 6h–k). The inhibition of acetate on the corresponding NLRP3 cascade, including NLRP3, pro IL-1β, cleaved IL-1β and cleaved caspase-1, was also abolished (Fig. 6l, m). However, the inhibitory effect of acetate on NF-κB signaling appeared to be unaffected by blocking ACSS2, as shown by western blotting (Supplementary Fig. 6).

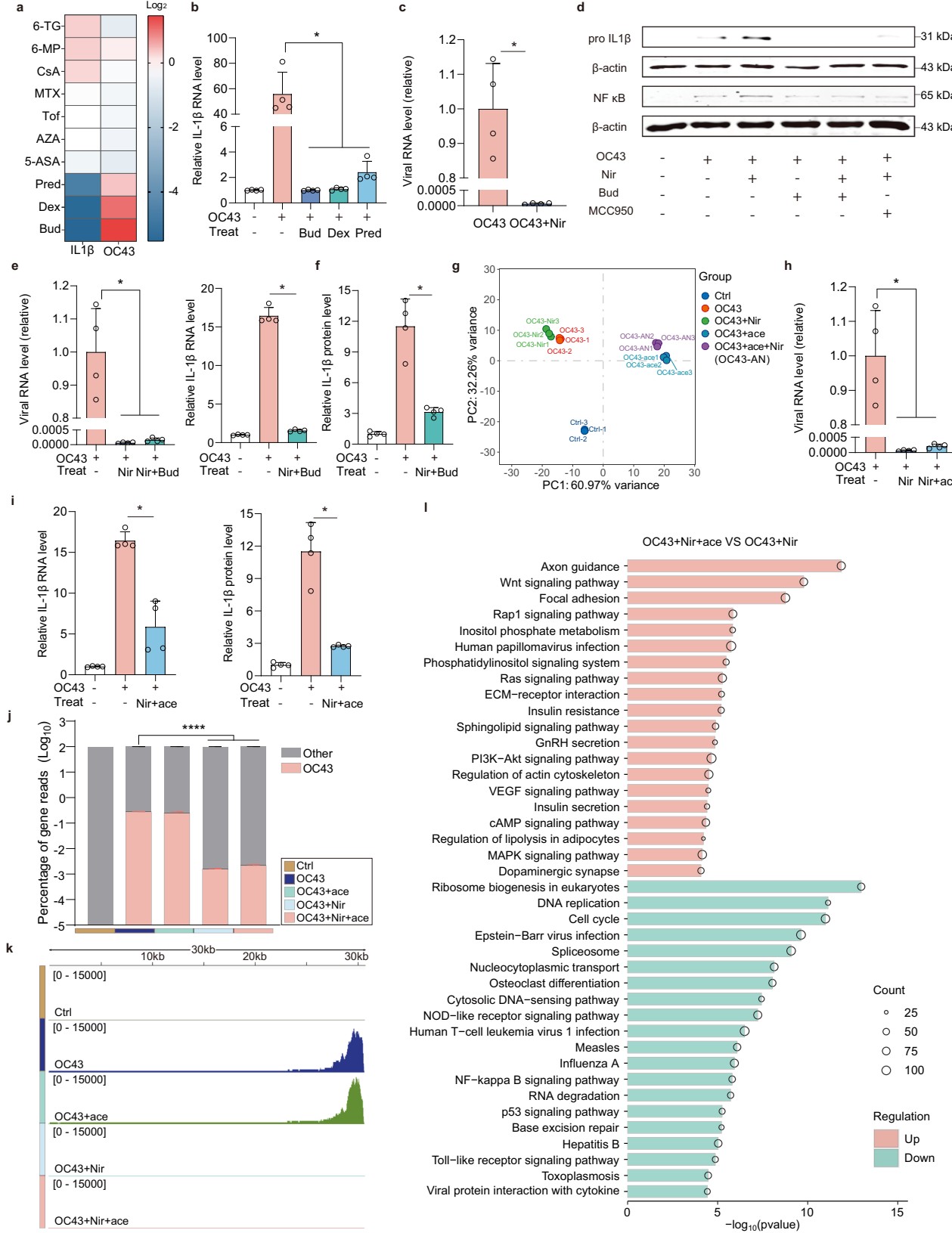

## Developing advanced treatment for enteric viral diseases: demonstrating a proof-of-concept in MaugOs

To identify a potent anti-inflammatory agent for treating enteric viral infections, we preliminarily screened 10 clinically used immunosuppressants in MaugOs challenged with OC43 virus for 36 hours. Notably, three classical steroids—prednisolone, dexamethasone and budesonide, dramatically attenuated *IL-1β* expression, albeit promoting viral replication (Fig. 7a). Further comparison of OC43-elicited *IL-1β* expression identified the strongest inhibitory efficacy of budesonide (Fig. 7b; Supplementary Fig. 7A). A similar anti-inflammatory effect exerted by budesonide was observed when inoculating MaugOs with rotavirus (Supplementary Fig. 7B). Next, we performed antiviral

**Fig. 7 | Devising combination treatment in MaugOs against enteric viral infections. a** Profiling the effect of a group of clinically used immunosuppressants on the expression of *IL-1β* and viral RNA in OC43-infected MaugOs at 36 hours post-treatment (*n* = 2). 6-TG: 6-Thioguanine; 6-MP: 6-mercaptopurine; CsA: Cyclosporin A; MTX: Methotrexate; Tof: Tofacitinib; AZA: Azathioprine; 5-ASA: 5-aminosalicylic acid; Pred: Prednisolone; Dex: Dexamethasone; Bud: Budesonide. **b** *IL-1β* RNA expression in OC43-infected MaugOs following 36-hours treatment of budesonide, dexamethasone, and prednisolone, respectively, (*n* = 4). **c** OC43 viral RNA level in MaugOs treated with 1 μM nirmatrelvir (Nir) for 36 hours (*n* = 4). **d** The effect of nirmatrelvir, budesonide and MCC950 (NLRP3 inhibitor) on the protein level of pro IL-1β and NF-κB at 36 hours post-treatment in OC43-infected MaugOs. **e**–**f** QRT-PCR quantification of OC43 viral RNA and IL-1β gene expression (**e**), and ELISA quantification of IL-1β production (**f**) in MaugOs following 36-hours combination treatment with 1 μM nirmatrelvir and 1 μM budesonide (*n* = 4). **g** PCA analysis of different MaugOs groups (*n* = 3). Control: uninfected MaugOs; OC43: OC43-infected MaugOs; OC43+Nir: 1 μM nirmatrelvir treatment in OC43-infected MaugOs; OC43+Acetate: 50 mM acetate treatment in OC43-infected MaugOs; OC43+Nir+Acetate: combination treatment of 1 μM nirmatrelvir and 50 mM acetate in OC43-infected MaugOs. **h**–**i** OC43 viral RNA level (**h**), IL-1β gene expression and protein production (**i**) in MaugOs following 36-hours combination treatment with 1 μM nirmatrelvir and 50 mM acetate (ace) (*n* = 4). **j** The percentages of mapped OC43 viral transcripts in different groups of MaugOs (*n* = 3). **k** Whole genome view of mapped transcripts by RNA-seq across the OC43 genome in different groups of MaugOs (*n* = 3). **l** Top 20 significantly regulated KEGG pathways in OC43-infected MaugOs upon combination treatment of nirmatrelvir and acetate, compared to nirmatrelvir treatment alone (*n* = 3). All data were presented as means of biological replicates ± SEM. Panel **b, c, e, f, h** and **i** use Mann–Whitney test with two tailed; **J** uses One-way ANOVA followed by Dunn's multiple comparison test (two-sided). *$p < 0.05$; **$p < 0.01$; ***$p < 0.001$; ****$p < 0.0001$.

treatment using nirmatrelvir, a potent agent known for its efficacy against SARS-CoV-2 and seasonal coronavirus. A complete inhibition to OC43 replication in MaugOs was achieved after 36 hours of nirmatrelvir treatment (Fig. 7c). Strikingly, the protein level of pro IL-1β and NF-κB activated by OC43 was further enhanced by nirmatrelvir treatment (Fig. 7d). This pro-inflammatory effect of nirmatrelvir was supported by KEGG analysis (Supplementary Fig. 7C).

To provide a proof-of-concept for devising combination treatment, we assessed the efficacy of combining budesonide and nirmatrelvir for treating coronavirus OC43 infection in MaugOs integrated THP-1 macrophages. Importantly, this combination resulted in the concurrent inhibition of viral replication and *IL-1β* expression (Fig. 7e). The anti-inflammatory impact was further confirmed by significantly reduced secretion of IL-1β (Fig. 7f). To broaden the therapeutic implication of acetate, we further devised combination treatment involving acetate and nirmatrelvir. PCA analysis revealed the distinct effects of monotherapy by nirmatrelvir or acetate, and their combination (Fig. 7g). Notably, combining acetate and nirmatrelvir significantly suppressed OC43 replication (Fig. 7h), and markedly decreased IL-1β expression and production (Fig. 7i). Consistently, a dramatic reduction of OC43 transcripts was observed by the combination treatment compared with untreated MaugOs (Fig. 7j). Characterizing the genomic location of mapped viral transcripts further demonstrated that the combination of acetate and nirmatrelvir effectively blocked the OC43 transcripts throughout the viral genome (Fig. 7k). In addition, compared to the antiviral monotherapy, KEGG analysis showed that anti-inflammatory pathways and intestinal physiology-associated pathways were upregulated, and pro-inflammatory-associated pathways were downregulated upon the combination treatment (Fig. 7l). Overall, the combination treatment of acetate with nirmatrelvir largely prevented the transcriptomic alteration by OC43 infection in MaugOs (Supplementary Fig. 7D, E).

Lastly, we validated these treatments employing SARS-CoV-2 as the disease modality. We showed that MaugOs supported the efficient propagation of SARS-CoV-2, which in turn triggered antiviral and inflammatory responses (Supplementary Fig. 7F-J). Similarly, nirmatrelvir treatment dramatically blocked SARS-CoV-2 replication but promoted the inflammatory response (Supplementary Fig. 7K, L). We next tested the combination treatment of nirmatrelvir with budesonide or acetate. As expected, SARS-CoV-2 replication and the inflammatory response were potently abolished by combination treatments (Supplementary Fig. 7M, N). Collectively, the combination treatment of nirmatrelvir with budesonide or acetate can potently inhibit SARS-CoV-2 infection and concurrent inflammatory response in MaugOs.

## Discussion
Historically, studying the pathogenesis of enteric viruses has primarily relied on 2D cultures of intestinal cell lines. However, these transformed cell lines harbor numerous genetic, epigenetic and functional alterations, and fail to replicate the cellular diversity and intricate structure of the intestine[26]. The advent of human intestinal organoids provides a novel platform for investigating enteric viral diseases[27]. These 3D-structured organoids, derived from LGR5-marked stem cells isolated from small intestine or colon, can better recapitulate the cellular heterogeneity and physiological environment of the intestinal epithelium[19,28,29]. Nevertheless, traditional intestinal organoids are lacking of immune cells, and thus hardly capture the essential host immune responses, in particular the inflammatory responses commonly seen during viral infections in patients. Research on developing complex in vitro models incorporating intestinal epithelium and immune components is emerging. For instance, intestinal organoids-derived 2D cell monolayers has been co-cultured with macrophages in a transwell system to evaluate barrier function upon bacterial infections[30]. Most recently, two experimental models integrating epithelial organoids and the immune cell compartment have been reported primarily for studying tissue development or immune responses in tumorigenesis[31,32]. However, the potential of these models for studying viral pathophysiology remains largely unexplored. In the current study, we engineered MaugOs through exquisitely integrating primary intestinal organoids with human macrophages. We further illustrated the application of MaugOs in studying viral life cycle, host antiviral and inflammatory responses, discovering biological implications of microbial metabolites, as well as developing advanced treatment strategies.

We prioritized echovirus, rotavirus and coronavirus as disease modalities in MaugOs. We demonstrated that MaugOs robustly supported the propagation of these viruses and more interestingly, we observed that viruses can be transmitted from intestinal organoids to macrophages, which resembles the natural infection process in the intestine where the virus first penetrates the epithelial barrier and subsequently interacts with immune cells[33]. In addition, the genome-wide transcriptomic analysis suggested active interactions between organoids and macrophages, as evidenced by the exclusive activation of specific genes and pathways in MaugOs. A meta-analysis has estimated that approximately 7% of patients infected with coronaviruses OC43 exhibit diarrhea[6]. Here we found that OC43 infection can elicit potent inflammatory responses in MaugOs via activating the NLRP3 inflammasome, which may partially explain the pathogenic mechanism of coronavirus OC43 within the intestine. Interestingly, we found that OC43 preferentially targeted enterocyte-differentiated organoids in our model, highlighting the potential of MaugOs incorporating differentiated organoids for investigating the cellular tropism of enteric viruses. Rotavirus remains the predominant cause of diarrheal fatalities among young children[34]. Prior insights into rotavirus pathophysiology are primarily derived from animal models, and suggests that rotavirus-induced inflammation is generally mild[35]. Similarly, our results revealed that rotavirus can trigger relatively mild inflammatory responses in MaugOs. Echoviruses are known to initiate the infection

through the intestine but rarely induce enteric pathogenesis[36]. Our findings indicated that echovirus 1 and echovirus 6 were not able to trigger an inflammatory response in MaugOs, despite the robust replication in macrophages. Therefore, we infer that productive infection may not be a trigger for initiating an inflammatory response in macrophages, at least in the case of echoviruses. We also recapitulated the infection of an early SARS-CoV-2 Omicron variant in MaugOs. In fact, the SARS-CoV-2 receptor, angiotensin-converting enzyme 2 (ACE2), is highly expressed in the intestine[37], which positions the intestine as a breeding ground for SARS-CoV-2 replication. Alarmingly, the latest dominant strain of SARS-CoV-2 has demonstrated an increased capacity for replicating in the gut[38]. Nevertheless, further investigations are warranted to determine whether the augmented replication of SARS-CoV-2 variant exacerbates intestinal inflammation, with MaugOs emerging as a promising model for such studies.

Bacterial components or products such as gram-negative bacterial toxin LPS, has been evidenced to mediate intestinal inflammation[39]. We here found that LPS can activate the NLRP3 inflammasome and lead to the release of pro-inflammatory cytokines in MaugOs. Our previous study identified NF-κB as an upstream event in hepatitis E virus-induced NLRP3 inflammasome activation[40]. Consolidating this finding, we demonstrated that both LPS and coronavirus OC43-induced NLRP3 inflammasome cascade were blocked by a NF-κB pharmaceutical inhibitor. Furthermore, severe enteric viral infections commonly alongside secondary bacterial infections, which largely contributes to the morbidity and mortality partly via exacerbating the pathological inflammation[41]. We recapitulated this scenario by co-stimulating the MaugOs with bacterial LPS and coronavirus OC43 and importantly, we effectively captured the synergistic inflammatory response, which underscores the sensitivity of MaugOs and relevance for studying complex enteric infection. Nevertheless, the diversity and abundance of cell types in MaugOs remain limited, which cannot fully represent the in vivo intestinal environment. Enhancing the cellular heterogeneity in MaugOs, for instance, incorporating additional cell types such as dendritic cells and stromal components (e.g., endothelial and smooth muscle cells), would enable to recapitulate the multidimensional aspects of enteric infections and further augment the translational value of the model.

Short-chain fatty acids (SCFAs) have been shown to involve a magnitude of host pathophysiological process, and intestinal immune response is particularly affected[42]. For instance, SCFAs butyrate has been implicated to facilitate the M2 macrophage polarization, thereby reducing intestinal inflammation[43]. Similarly, our genome-wide transcriptomic analysis revealed that acetate treatment induced a macrophage phenotype shift from M1 to M2 in MaugOs. In line with previous studies that SCFAs can suppress the activation of NF-kB and NLRP3, we here found that acetate dramatically blocked the expression of NF-kB and NLRP3 inflammasome in MaugOs. From the mechanism-of-action, SCFAs primarily function through surface-expressed FFAR or cytoplasmic ACSS2[44]. Our study demonstrated that acetate profoundly prevented the inflammatory response in MaugOs partially through FFAR2 and ACSS2 signaling. Notably, we discovered that FFAR2 was pivotal for acetate to inhibit NF-kB and NLRP3 inflammasome, whereas ACSS2 specifically involved the inhibition of acetate to NLRP3 but not NF-kB. Interestingly, acetate has been shown to prevent influenza A virus infection by enhancing the host antiviral response in mice model[45], whereas butyrate has been indicated to promote virus infection (e.g., influenza virus, reovirus, HIV-1) by suppressing antiviral response in multiple cell lines[46]. In contrast, we found that acetate had a minimal-to-mild inhibition to coronavirus OC43 in MaugOs, due to the disparate influence to viral infection in macrophages and intestinal organoids.

Despite significant advances in modern medicine for the prevention and treatment of viral diseases, specific treatments for enteric viral infections remain limited[47,48]. To advance the treatment strategy for enteric viral infections, we provided a proof-of-concept by combination treatment of nirmatrelvir with budesonide, which effectively inhibited viral replication and the resulting inflammatory responses in MaugOs challenged with seasonal coronavirus OC43 or pandemic SARS-CoV-2. Particularly, the drug concentrations we used for treating viral infections in MaugOs were within clinically achievable ranges. We recommend selecting clinically relevant concentrations as the first-option when testing treatments in models like MaugOs. This strategy would largely enhance the translational potential of the developed therapies for clinical practice. In addition, our study identified the dramatic anti-inflammatory effects of acetate, even in the co-infection scenario involving thebacterial component and virus. We further demonstrated the robust antiviral and anti-inflammatory efficacy of combining acetate with nirmatrelvir. Genome-wide transcriptome analysis revealed profound rewiring of host gene transcription by coronavirus OC43 infection in MaugOs, whereas combination treatment of acetate and nirmatrelvir largely impeded this process. These findings collectively implicate the feasibility of combining acetate with antiviral regimens for treating infections and inflammatory diseases, although further in vivo studies or clinical trials are required to solidify these results.

In summary, this study established macrophage-augmented intestinal organoids and this model effectively recapitulated complex virus-host interactions induced by distinct enteric viral infections. We discovered the pleiotropic effects of acetate regarding the viral infection and inflammatory response. Furthermore, we devised combination treatments that potently inhibited both viral infection and inflammatory response.

## Methods

### Ethics Statement

The use of human intestinal tissue for research purposes in this study was approved by the Medical Ethical Council of the Erasmus MC, and informed consent was given (MEC-2021-0432; MEC-2023-0629).

### Regents

Immunosuppressants and the antiviral drug, including 6-mercaptopurine, Methotrexate, tofacitinib, Cyclosporin A, budesonide, 5-aminosalicylic acid, azathioprine, dexamethasone, prednisolone, 6-Thioguanine and nirmatrelvir were dissolved in dimethyl sulfoxide (DMSO) with a stock concentration of 100 mM. LPS was dissolved in DMSO with a stock concentration of 1 mg/mL. Short-chain-fatty acids including acetate, butyrate and propionate were dissolved in DMEM medium with a stock concentration of 1 M. The detailed information of these regents was provided in supplementary Table 4.

### Cell lines and viruses

Multiple cell lines including human hepatoma cell line Huh7, monkey kidney cell line Vero-E6, adenocarcinomic human alveolar basal epithelial cell line A549 and African green monkey kidney cell line MA-104 were cultured with Dulbecco's modified Eagle medium (DMEM) (Lonza Biowhittaker) supplemented with 10% (vol/vol) heat-inactivated fetal calf serum (FCS, Sigma–Aldrich), 100 IU/mL penicillin, and 100 mg/mL streptomycin (Gibco). The human lung cancer cell line Calu-3 was cultured in advanced DMEM/F12 supplemented with 1% (vol/vol) GlutaMAXTM Supplement (Gibco), 10 mM HEPES (Life Technologies), 100 IU/mL Penicillin and 100 mg/mL streptomycin (Gibco) and 10% (vol/vol) heat-inactivated fetal calf serum (FCS, Sigma–Aldrich). The human monocytic cell lines THP-1 were cultured in RPMI 1640 Medium (Thermo Fisher) complemented with 10% (v/v) inactivated Fetal Bovine Serum with 100 IU/mL penicillin and 100 mg/mL streptomycin. EV1 (GenBank: AF029859) and EV6 (GenBank: JQ929657) were amplified in A549 cells. OC43 virus (GenBank: AY585228) was amplified in Huh7 cells. Rotavirus (SA11 strain, GenBank: X16830) was amplified in MA-104 cells. SARS-CoV-2 (GenBank: MT270101) was amplified in Calu-3

cells. Cell lines used in this study were analyzed by genotyping and confirmed to be mycoplasma negative.

## Intestinal organoids culture

Human primary intestinal organoids were isolated from intestinal biopsies and cultured as our previously established protocol[49]. These organoids were cultured in organoid expansion medium (OEM), based on advanced DMEM/F12 (Invitrogen), supplemented with 1% penicillin/ streptomycin (Life Technologies), 10 mM HEPES, 1xGlutamax, 1 mM N2, 1 mM B27 (all from Invitrogen), 1 μM N-acetylcysteine (Sigma) and the following growth factors: 50 ng/L mouse epidermal growth factor (mEGF), 50% Wnt3a-conditioned medium (WCM) and 10% noggin-conditioned medium (NCM), 20% Rspo1-conditioned medium, 10 μM nicotinamide (Sigma), 10 nM gastrin (Sigma), 500 nM A83–01 (Tocris) and 10 μM SB202190 (Sigma). The medium was refreshed every 2-3 days, and organoids were passaged 1:3 every 5–7 days.

## Differentiation of organoids

Organoids were further differentiated toward different cell types by using different differentiation medium. Briefly, organoids were cultured in OEM supplemented with 2 μM IWP-2 (Sigma) for enterocytes differentiation; OEM supplemented with 2 μM IWP-2 and 10 μM DAPT (MedChemExpress) for goblet cells differentiation; OEM supplemented with 2 μM IWP-2, 10 μM DAPT and 5 μM Gefitinib (Sigma) for enteroendocrine cell differentiation. The differentiation medium was refreshed every 2-3 days and expected cell types were achieved after 5 days differentiation culture.

## Macrophage differentiation from THP-1 monocytic cells

To generate non-activated M0 macrophages, human monocytic THP-1 cells were cultured in RPMI 1640 medium, supplemented with 30 ng/mL of phorbol 12-myristate 13-acetate (PMA) at 37 °C for 48 hours.

## Isolation and differentiation of primary macrophages

Blood monocyte-derived primary macrophages were generated following our previously described method[40]. In short, peripheral blood mononuclear cells (PBMCs) were isolated from healthy donors (Sanquin, The Netherlands) using Ficoll density gradient centrifugation. Then monocytes were specifically separated through plastic adherence in Iscove's modified Dulbecco's medium (IMDM) supplemented with 2% human serum. The isolated monocytes were cultured in IMDM (with ultraglutamine), supplemented with 8% (v/v) inactivated FCS, 1% penicillin/streptomycin (Gibco), and 50 ng/mL of macrophage colony-stimulating factor (M-CSF), for a period of 7 days to generate mature macrophages.

## Differentiation of induced pluripotent stem cells-derived macrophages

Human induced pluripotent stem cells (iPSCs) were used to generate monocytes following our previously established methods[50]. The Allen Cell Collection line AICS-0061 (generated from fibroblasts, https://hpscreg.eu/cell-line/UCSFi001-A-28) with mEGFP insertion site at HIST1H2BJ was obtained from Coriell Institute for Medical Research[51]. For cell adherence, 0.5 mL of fetal bovine serum (FBS) was added to each well of a 12-well plate and incubated overnight at 37 °C. Monocytes were thawed in IF9S medium supplemented with RevitaCell (final concentration 0.5 X). Next, 0.75 million monocytes were plated onto the FBS coated well and cultured in IMDM/F12 with nine supplements (IF9S), based on IMDM supplemented (Invitrogen) with 50% F12 Nutrient Mix (Invitrogen), 10 ng/mL Polyvinyl alcohol (Sigma), 0.1% Chemically Defined Lipids (Invitrogen), 2% Insulin-Transferrin-Selenium-X (Invitrogen), 40 μL/L Monothioglycerol solution (Sigma), 64 mg/L L-Ascorbic Acid 2-Phosphate (Sigma), 2 mM GlutaMAX (Invitrogen), 1% non-essential amino acids (Invitrogen) and 0.5% Penicillin-Streptomycin. The IF9S medium supplemented 80 ng/mL M-CSF

(Sigma) was refreshed every 1 to 2 days, and the differentiation process was continued for a period of 7 days.

## Flow cytometric analysis

THP-1, PBMCs and iPSCs-differentiated macrophages were further analyzed by determining the expression of surface and intracellular markers. Primary antibodies used in this study are as follows: CD14-eFluor450 (1:50), CD32-PE (1:200), CD16-FITC (1:50), CD68-BV785 (1:200), CD80-PECy7 (1:25), HLA DR-BV605 (1:200), CD14-BV711 (1:50) and CD45-APC-Fire750 (1:30). Dead cells were excluded using live/ dead stain eFluor506 (Thermofischer Scientific, Bleiswijk, the Netherlands). To reduce unspecific binding of the antibodies, cells were blocked with mouse serum, human Fc-block (BDbioscience, Erembodegem, Belgium) and true-stain Monocyte Blocker (Biolegend, Amsterdam, the Netherlands). Cell surface staining with fluorochrome-conjugated antibodies was performed in the dark at 4 °C for 15 min, after which cells were fixed and permeabilized using the FoxP3 staining buffer set (Thermofischer Scientific, Bleiswijk, the Netherlands) and stained for intracellular antigens (CD68). Cells were measured on a Symphony A3 flow cytometer (BD Biosciences, San Diego, USA) and analyzed using FlowJo software version 10.10.0 (Becton Dickinson). Appropriate isotype control antibodies were used for gating purposes.

## Establishment and characterization of macrophage-augmented organoids

When organoids were over 75% confluence in Matrigel and each single organoid is approximately 200 μm of diameter, these organoids were then proceeded to integrating with macrophages. Briefly, organoids were first washed in cold Advanced DMEM/F12 and centrifuged at 300 g for 5 minutes to remove basal matrix. Subsequently, organoids were mechanically dissociated into small fragments. THP-1 macrophages, primary and iPSCs-derived macrophages were then respectively mix with fragmented organoids, at a ratio of 100 organoids to 10^5 macrophages. Diluted Matrigel from the ratio of 1:5 to 1:20 were pre-supplemented in culture plate for providing a supporting basement to organoids and cells. The mixture of macrophages and fragmented organoids were then seeded on the surface of diluted Matrigel. As two control groups, organoids and macrophages were cultured separately. Afterwards, MaugOs, organoids, and macrophages were treated with 1 μg/mL LPS to assess the response to inflammatory stimulus. For macrophages derived from THP-1 cells and PBMCs, they were pre-labeled with 5 μM CFSE (Thermo Fisher Scientific, Cat. C34554) in PBS at 37 °C for 30 minutes, then centrifuged and washed with macrophage growth medium to remove excess dye. Macrophages differentiated from iPSCs-derived monocytes expressed H2B-mEGFP protein. These macrophages were then integrated with organoids to generate MaugOs. EVOS and LEICA SP5 confocal microscopy were used for visualization. For the characterization and application of the MaugOs model, organoids derived from three different donors were used to validate key findings, while one representative donor was employed for the majority of experiments in this study.

## Virus inoculation

Organoids were mechanically fragmented and then exposed to viral particles for 2 hours at 37 °C for rotavirus, EV1 and EV6, and at 33 °C for OC43 virus. Each organoid was inoculated with approximately 4 × 10^4 PFU of OC43 virus, 10^3.8 PFU of rotavirus, 2 × 10^4 PFU of echovirus, respectively, (rotavirus was pre-activated by 0.05% trypsin for 10 minutes at 37 °C). To increase the infection efficacy, organoids and virus mixture were resuspended every 30 minutes throughout the inoculation period. Subsequently, fragmented organoids underwent centrifugation at 300 g for 5 minutes at 4 °C, and the supernatant was discarded. Then organoids were thoroughly washed three times with advanced DMEM/F12 to remove residual viruses. After infection, the organoids were integrated with macrophages in Matrigel-coated wells.

Non-infected organoids were integrated with macrophages under identical conditions as controls.

For the infection to macrophages, macrophages were inoculated with different virus particles (MOI = 0.1) for 2 h at 37 °C (OC43 was exposed for 2 hours at 33 °C). Infected macrophages were then washed by PBS for 3 times to thoroughly remove unabsorbed virus. Finally, macrophages were maintained in RPMI 1640 medium supplemented with 4% FBS at 37 °C with 5% $CO_2$.

### Antiviral and anti-inflammatory treatment

Ten clinically used immunosuppressants (1 μM) were administered to LPS-stimulated THP-1-derived macrophages for 36 hours to assess their anti-inflammatory efficacy. As a proof-of-concept, acetate (50 μM) was applied to evaluate the effect of short-chain fatty acids. For antiviral treatment, nirmatrelvir (1 μM) was used to treat OC43- and SARS-CoV-2-infected MaugOs for 36 hours. Independent organoids or MaugOs cultures derived from a separate well was regarded as one biological replicate.

### Immunofluorescence staining and confocal imaging

To visualize the integral MaugOs by immunofluorescence staining, macrophage and fragmented organoids were seeded into μ-Slide 8 Well (ibidi GmbH, 80826) pre-coated with diluted Matrigel (1:10). After 36 hours postseeding, MaugOs were fixed by exposing the cells to 4% paraformaldehyde (PFA) for 15 min. Subsequently, the sample-containing μ-Slide wells were gently rinsed 3 times with PBS, followed by permeabilizing with PBS containing 0.2% (vol/vol) Triton X-100 for 10 min. Then the μ-Slide wells were twice rinsed with PBS for 5 min, followed by incubation with blocking solution (5% donkey serum, 1% bovine serum albumin, 0.2% Triton X-100 in PBS) at room temperature for 1 hour. Next, primary antibodies diluted in blocking solution were incubated with samples at 4 °C overnight. Primary antibodies used in this study are as follows: anti-EpCAM (1:500, rabbit), anti-dsRNA antibody (1:500, mouse), anti-CD68 antibody (1:500, mouse), anti-Villin (1:100, mouse), anti-MuC2 (1:100, mouse), anti-CHGA (1:100, mouse), anti-SARS-CoV-2-nucleocapsid protein (1:500, mouse). Sample-containing μ-Slide wells were then washed 3 times for 5 min each in PBS prior to 1 hour incubation with 1:1000 dilutions of the secondary antibodies, including anti-mouse IgG (H + L, Alexa Fluor® 594) and the anti-rabbit IgG (H + L, Alexa Fluor® 488). Nuclei were stained with DAPI (4, 6-diamidino-2-phenylindole; Invitrogen). At last, stained samples were visualized using a Leica SP5 confocal microscope with a 40× oil immersion objective to analyze the stained cellular structures.

### Immunohistochemistry (IHC) staining

MaugOs were fixed in 2% paraformaldehyde for 20 minutes, followed by paraffin embedding and sectioning into 4 μm-thick slides. Immunohistochemistry (IHC) procedures were performed to detect antigens of CD68 for macrophages. Paraffin sections mounted onto 3-amino-propyltriethoxysilane-coated glass slides (Epredia, USA) were deparaffinized in xylene, rehydrated, and rinsed in phosphate-buffered saline (PBS), pH 7.4. For antigen retrieval, sections were treated in a 10 mM citrate buffer (pH 6.0). Endogenous peroxidase was inactivated by covering the sections with 3% $H_2O_2$. Sections were then incubated with the antibody anti-CD68 diluted at 1:500 in a humidified chamber at 37 °C for 90 minutes. We used UltraVision LP Detection System, polymer –horseradish peroxidase (HRP) (VectorLabs, USA) staining was completed using the chromogen solution, 3,3'- diamino-benzidin (DAB).

### Time-lapse microscopy

The organoids fragments and CFSE pre-labeled THP-1 macrophages were cultured in μ-Slide 8 Well (ibidi GmbH, 80826) after assembling. The cell cultures were then subjected to imaging employing the sophisticated Opera Phenix system by PerkinElmer, which was equipped with a 10 × air objective (NA 0.3). To capture the dynamic process of MaugOs formation, a small z-stack was acquired at 30-minute intervals over an 18-hour period, ensuring optimal optical focus at multiple points throughout the experiment.

### Genome-wide RNA sequencing and data analysis

Organoids infected with EV and OC43 virus, respectively, were integrated with THP-1 macrophages for 36 hours to generate EV and OC43 infected MaugOs. THP-1 macrophages were inoculated with EV and OC43 and cultured for 36 hours as two infected THP-1 groups. Similarly, organoids were inoculated with EV and OC43, respectively, and cultured for 36 hours as two infected organoids groups. In parallel, non-infected organoids, macrophages, and MaugOs were cultured under same conditions as negative controls. Total RNA was isolated using the MachereyNagel NucleoSpin RNA II Kit (Bioke, Netherlands) and quantified with the Bioanalyzer RNA 6000 Picochip. Afterwards, RNA sequencing was conducted by Novogene using a paired-end 150 bp (PE 150) sequencing strategy.

To map the viral genome from transcriptomic data, quality control of raw sequencing reads was performed using Fastp (version 0.23.2) to remove low-quality reads and adapter sequences. The filtered reads were then aligned to the reference genome (OC43 ATCC VR-759, GenBank Accession No. AY585228; Echovirus 1 ATCC VR-1038, GenBank Accession No. AF029859) using Bowtie2 (version 2.4.4). Format conversion and sorting were performed with SAMtools (version 0.1.19). Read coverage across the genome was assessed using Bedtools (version 2.31.0), and visualization of viral gene mapping was performed with Integrative Genomics Viewer (version 2.1.2). Differential gene expression analysis was performed in R (version 3.6.0) using the DESeq2 package. For transcript-level differential expression analysis, the edgeR package was used with Benjamini−Hochberg correction for multiple testing, applying a false discovery rate (FDR) threshold of $10^{-5}$. Functional enrichment analysis of differentially expressed genes was conducted using clusterProfiler package for KEGG pathway annotation. Gene Set Enrichment Analysis (GSEA) was performed using clusterProfiler and fgsea. Overlapping differentially expressed genes among experimental groups were visualized using ggVennDiagram. Volcano plots were generated using the EnhancedVolcano package, while heatmaps were created using GraphPad Prism 8. To enhance interpretability in heatmaps, Z-score transformation was applied to gene expression data. Additional statistical analyses and graphical visualizations were conducted using ggplot2.

### Macrophage-augmented organoids in transwell culture system

Organoids were inoculated with OC43 virus for 2 hours at 33 °C, then embedded in Matrigel and seeded in the lower compartment of the trans-well culture system. Approximately 10^5 THP-1 macrophages were seeded in the upper trans-well inserts (Corning BV, 4μm/8μm), and cultured for 1 hour and 48 hours. Control groups followed the same steps without viral exposure. RNA samples were collected from both compartments simultaneously.

For the cell migration assay in the transwell system, the insert was fixed using 4% paraformaldehyde for 15 minutes. After fixation, aspirate the solution and transfer the insert to a well containing 800 μL of 0.1% crystal violet staining solution for 15 minutes. Subsequently remove unbound crystal violet and rinse gently with PBS. Aspirate the liquid and wipe the cell surface on the membrane. Then peel off the membrane and placing it bottom-side up for drying, and transfer it to a microscope slide covered with neutral gum. At last, count the purple-stained positive cells in five randomly selected fields (top, bottom, center, left, right).

### Enzyme-linked immunosorbent assay (ELISA)

Released cytokines in the culture supernatant of MaugOs were determined using commercial Enzyme-Linked Immunosorbent Assay (ELISA) Kits. The concentrations of interleukin-1 beta (IL-1β),

interleukin-6 (IL-6), interleukin-8 (IL-8) and tumor necrosis factor alpha (TNF-α) were measured by the IL-1β Kit (ThermoFisher Scientific, Cat# 88-7261-88), IL-6 Kit (ThermoFisher Scientific, Cat# 88-7066-22), IL-8 Kit (ThermoFisher Scientific, Cat# 88-8086-88) and TNF-α Kit (ThermoFisher Scientific, Cat# 88-7346-88) respectively. The release of lactate dehydrogenase (LDH) into cell culture supernatant was assessed using the CytoTox 96® Non-Radioactive Cytotoxicity Assay Kit (Promega).

## AlamarBlue assay
Culture supernatant was discarded, and the organoids or macrophages were incubated with a 1:20 dilution of AlamarBlue regent (Invitrogen, DAL1100) in culture medium for 2 hours at 37 °C. Subsequently, 100 μL medium was collected to assess cell metabolic activity, with each sample being measured in duplicate. Absorbance measurements were obtained using a fluorescence plate reader (CytoFluor Series 4000, PerSeptive Borganoidsystems) at an excitation wavelength of 530/25 nm and an emission wavelength of 590/35 nm (λExc 530 nm/λEm 590 nm).

## QRT-PCR quantification of gene expression
Total RNA was extracted using the Macherey-Nagel NucleoSpin RNA II Kit (Bioke, Leiden, Netherlands). The concentration and purity were measured using the Nanodrop ND-1000 (Wilmington, DE, USA). Gene expression levels were quantified by SYBR Green–based qRT-PCR using the Applied Borganoidsystems SYBR Green PCR Master Mix (Thermo Fisher Scientific Life Sciences) with the StepOnePlus System (Thermo Fisher Scientific Life Sciences). The housekeeping gene used for normalization was glyceraldehyde 3-phosphate dehydrogenase (GAPDH). Relative gene expression was normalized to GAPDH using the $2^{-\Delta\Delta CT}$ method ($\Delta\Delta CT = \Delta CT\_sample - \Delta CT\_control$). Each qRT-PCR experiment included template control and reverse transcriptase control. OC43 copy numbers were detected by primers 'Forward- AGCAACCAGGCTGATGTCAATACC; Reverse- AGCAGACCTTCCTGAGCCTTCAAT', and calculated by previously generated formula '$y = -0.2794x + 10.974$'[52]. All other primers used in this study are listed in Supplementary Table 5.

## Immunoblot analysis
Sample lysates were heated at 95 °C for 5 minutes. Proteins were subjected to a 10% or 15% sodium dodecyl sulfate polyacrylamide gel (SDS-PAGE), separated at 90 V for 120 minutes, and electrophoretically transferred onto a PVDF membrane (pore size: 0.45 μm; Thermo Fisher Scientific Life Sciences) for 120 minutes with an electric current of 250 mA. Following the transfer, the membrane was blocked with blocking buffer (LiCor Borganoidsciences). Primary antibodies were applied and allowed to incubate overnight at 4 °C. Primary antibodies used in the study are as follows: anti-NLRP3 antibody (1:1000, Rabbit), anti-IL-1β (1:1000, rabbit), anti-Cleaved-IL-1β (1:1000, rabbit), anti-NF-κB (1:1000, rabbit), anti-β-actin (1:1000, mouse) and anti-villin (1:1000, mouse). Subsequently, the membrane underwent three washes, followed by a 1-hour incubation with anti-rabbit or anti-mouse IRDye-conjugated secondary antibodies (1:5000; Li-Cor Borganoidsciences) at room temperature. After three additional washes, protein bands were visualized using Odyssey 3.0 software.

## Statistics and reproducibility
For representative images and data with statistical analysis, each experiment was either repeated independently at least three times or included a minimum of three biological replicates. Statistical analysis was performed using GraphPad Prism8 statistics software (GraphPad, San Diego, USA). All data are presented as mean ± standard error of the mean (s.e.m.). *T* test and One-way ANOVA were used for statistical analysis. Asterisks indicated the degree of significant differences compared with the controls (*$p < 0.05$; **$p < 0.01$; ***$p < 0.001$;

****$p < 0.0001$). All significance analyses and other relevant information for data comparison are specified in sthe ource data.

## Reporting summary
Further information on research design is available in the Nature Portfolio Reporting Summary linked to this article.

## Data availability
This study did not generate new unique reagents. Genome-wide RNA-sequencing data generated in this study are publicly available at the NCBI database under accession code GSE295767. Data that supports the findings of this study are available in the main text, supplementary information and source data. Source data are provided with this paper.

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

## Acknowledgements

The authors would like to thank Dr. Gert-Jan Kremers (Erasmus Optical Imaging Center) for assisting with the time-lapse imaging. This work was supported by a Young Investigator Grant from Erasmus MC-University Medical Center and a pandemic preparedness grant (No. 10710032310013) from The Netherlands Organisation for Health Research and Development (ZonMw) to P.L.; the COVID-19 Pro-gramme (No. 50-56300-98-2201) and a VIDI grant (No. 91719300) from ZonMw to Q.P.; the LymphChip project with project number NWAORC 2019 1292.19.019 of the NWA research program and "Research on Routes by Consortia (ORC)" funded by the Netherlands Organisation for Scientific Research (NWO)" to V.O. The Allen Cell Collection, available from Coriell Institute for Medical Research, provided materials.

## Author contributions

G.G.X., Q.W.P., and P.F.L. conceived the study. G.G.X., J.R.Z., K.L., Y.N.W., and D.M.O. designed the experiments. G.G.X. and F.Q. per-formed data analysis and visualization. T.T., F.V.D.H., and V.V.O. pre-pared iPSC-derived monocytes. I.A., A.C.D.V., J.J.L., and M.J.C.B. cultured intestinal organoids. D.E.K. propagated echovirus. P.P.C.B. performed flow cytometry analysis. G.G.X. and P.F.L. wrote and revised the manuscript. S.J.J., H.L.A.J., W.S.W., and M.P.P. reviewed the manu-script. P.F.L. supervised the project and secured funding.

## Competing interests

The authors declare no competing interests.

## Additional information

[1]Department of Preventive Veterinary Medicine, College of Veterinary Medicine, Shandong Agricultural University, Taian, Shandong 271018, China. [2]Department of Gastroenterology and Hepatology, Erasmus MC-University Medical Center, Rotterdam, the Netherlands. [3]Precision Medicine Translational Research Center, West China Hospital, Sichuan University, Chengdu 610041, China. [4]Department of Surgery, Erasmus MC Transplant Institute, University Medical Center Rotterdam, Rotterdam, the Netherlands. [5]Department of Anatomy and Embryology, Leiden University Medical Center, Leiden, the Netherlands. [6]Department of Pathogen Biology and Immunology, Jiangsu Key Laboratory of Immunity and Metabolism, Jiangsu International Laboratory of Immunity and Metabolism, Xuzhou Medical University, Xuzhou 221004, China. [7]Department of Clinical and Molecular Medicine (IKOM), Norwegian University of Science and Technology, 7028 Trondheim, Norway. [8]Toronto Centre for Liver Disease, Toronto General Hospital, University Health Network, Toronto, ON, Canada. [9]These authors contributed equally: Guige Xu, Jiangrong Zhou, Kuan Liu. ✉e-mail: p.li@erasmusmc.nl

