## [Peer Review file · Nature Communications]

Macrophage-augmented intestinal organoids model virus-host interactions in enteric viral diseases and facilitate therapeutic development

Corresponding Author: Dr Pengfei Li

Version 0:

Reviewer comments:

Reviewer #1

(Remarks to the Author)

The reviewer would like to begin by commending the authors on their extensive work. The manuscript describes the development, optimization, and characterization of a human in vitro intestinal model augmented with macrophages. The authors successfully demonstrate the applicability of this model in facilitating infection with human viruses, a commendable achievement and a significant step forward in the effective study of human viral-host interactions.

Notably, the authors demonstrate the feasibility of generating macrophages from various sources, including established human cell lines (THP-1), induced pluripotent stem cells (iPSCs), and blood monocytes. They also show that both iPSC- and blood-derived macrophages respond to inflammatory stimuli, enabling donor-matched models to be produced. Furthermore, the study focuses on clinically relevant human viruses requiring innovative treatment strategies. The authors emphasize the dual approach of identifying effective antiviral treatments while also examining the inflammatory responses that can exacerbate disease pathology in certain viral infections. This integrated approach is particularly valuable. The reviewer has several comments, categorized as minor and major, to aid the authors in further improving their manuscript.

Major Comments

Reproducibility, donor variation and virus characteristics.

The manuscript contains an extensive amount of data in figures with subfigures and supplementary figures. In the legend, the amount of replicates is stated for each experiment. However, the Methods section lacks information about what these replicates are exactly. Are these experiments repeats of organoids from the same donor? Different donors? What is the origin of the donors? Virus infections in organoids may be donor dependent, and host responses are most certainly donor dependent. This issue is not addressed in the manuscript.

The same lack of detail is observed in the description of the viruses studied here. The justification for the choice of viruses is unsatisfactory (there are many enteroviruses, why echovirus 1 and 6? Are these the main gastro-intestinal enteroviruses? Or are they the most common? Or the most related to serious disease?). There is no mentioning of the origin of the viruses, what strains and/or accession numbers, are these clinical isolates or lab-adapted strains? How were these viruses cultured before infecting the organoids? Fig 2: the relative viral RNA levels, relative to what? How is this calculated? Fig S3: here the read out is log viral copies /mg RNA, with very high copies already at day 0, was this also seen in the other replication experiments?

Physiological Relevance of Models:

The authors discuss the physiological relevance of their model and critique the lack of heterogeneity in current models. However, the methods indicate that their model is differentiated to express specific cell types—either enterocytes, enteroendocrine cells, or goblet cells—under different conditions. While this targeted differentiation is useful for studying cellular tropism, it does not fully recapitulate the heterogeneity of the human intestinal environment.

A more physiologically representative model, incorporating multiple cell types simultaneously, could provide deeper insights into intercellular communication and downstream effects of viral infections. For instance, the activation of PPAR γ , a pathway implicated in various viral infections, often involves localized effects that extend beyond the initially infected cells. The authors might consider discussing whether such heterogeneity would enhance the translational value of their model.

Novelty of Findings:

The authors claim their model to be novel. However, co-cultures of gut organoid models with macrophages have been described by other groups (already in 2017 by Noell et al., Scientific Reports). References to these studies are lacking and should be included.

The authors highlight their model's ability to elucidate cellular tropism for coronaviruses, emphasizing enterocytes as target cells. However, this observation is already well-established in literature. If the goal was to validate the model for studying cellular tropism, this should be explicitly stated. Otherwise, the current framing may imply novelty where it is not warranted.

Compound Selection and Dosing Information:

In Figures 4 and 6, the authors mention the compounds used (e.g., acetate, nirmatrelvir, and budesonide) but only provide concentration details for a subset of these. Additional information about origin (company where purchased), what solution buffer is used to dissolve the compounds, dosages, treatment frequency, and duration for all compounds would improve the reproducibility and interpretation of the results.

The data in Fig 4 and 6 are not convincing as they lack the control of the dissolvent buffer. DMSO or ethanol-based buffers could have an effect on the cells, therefore it should be proven that the effects the authors claim are in fact the effect of the SCFA or the antiviral compounds and not just due to the composition of the buffer.

Furthermore, a discussion of how the concentrations of the antivirals relate to clinically achievable dosages would be valuable, as clinical translatability is critical in such studies.

Rationale for Compound Selection:

The rationale behind the selection of compounds is not fully discussed. Were these compounds chosen as proof-of-concept agents for model validation, or are they believed to have clinical relevance? If the latter, the authors should discuss how these compounds could be implemented in clinical settings and whether their model could provide insights into their safety and efficacy. Addressing these questions would enhance the manuscript's translational significance.

Minor Comments

Ethical statement:

The authors do not report on the ethics and ethical considerations in the manuscript and this should be addressed.

Macrophage Characterization:

In the methods section, the authors state that macrophages were differentiated and characterized using monocyte and macrophage markers but do not specify which markers were used or their dilutions. Providing these details would enhance reproducibility.

Reference Gene Stability:

The authors mention the use of GAPDH as a reference gene in their qPCR analyses. Did the authors evaluate the stability of GAPDH expression in organoids and macrophages following viral infection? Viral infections can alter host gene expression, potentially affecting the suitability of certain reference genes and introducing biases into the data. The following references may provide valuable context:

<https://journals.asm.org/doi/full/10.1128/spectrum.01656-22>

<https://bmcbgenomics.biomedcentral.com/articles/10.1186/1471-2164-12-156>

Origin of antibodies

There is no mentioning of the origin of the antibodies used for staining in the Method section. Such information would help the reproducibility of the data by other groups.

Figure S1F (Viability Data):

Could the authors clarify the y-axis of this graph? Providing the units and explaining how the data should be interpreted would improve clarity.

Figure S1I (IL-33 Quantification):

The authors state that IL-33 was quantified, but this is not shown in Figure S1I. Instead, TNF α is displayed. Please clarify this inconsistency.

References

The reviewer noticed that the referencing is not always accurate. For example, ref 4 is a very specific study about characteristics of echovirus 1, while the authors make a statement about the enterovirus as a large group that affects many people worldwide.

Furthermore, it strikes me that there seems to be a substantial amount (>10%) of referring to the work of the authors (auto-referencing), also for general statements (for example, line 55 ref 8).

Final Remarks

While the reviewer appreciates the nature of the work in this manuscript, the overall conclusion is that the manuscript is not convincingly sound as some necessary methodological details are missing in the enormous amount of information provided. While the application of the model is quite exciting, the model developed here in itself is not novel. The suggested clarifications and additional details would further strengthen the manuscript and ensure its findings have the highest possible impact.

Reviewer #2

(Remarks to the Author)

This study by Xu et al. aimed to develop an in vitro model of viral enteric infections by integrating macrophages into

intestinal organoids. The authors showed that human macrophage-augmented intestinal organoids, called MaudOs, support replication of a panel of enteric viruses and produce virus-specific inflammatory responses. They characterize MaudOs through imaging and transcriptomics, and also test the impact of short chain fatty acids (SCFAs) and the antiviral nirmatrelvir. Based on the collective evidence, the authors conclude that MaudOs can be used to better recreate the human gut environment for the purpose of investigating virus-host interactions.

The cross comparison between different stimuli and viruses was much appreciated and interesting. I also agree with the conclusion that the MaudOs faithfully recreate the function of the individual cell types represented in the culture. As the authors correctly state, the lack of immune cells is a major limitation of intestinal organoid models, although some new papers (including citation #34) are making advances that include macrophages, lymphocytes, and other cell types. Given that these newer models are still preliminary, I see tremendous value in developing the type of approach featured in this current study, even if it is limited by only focusing on macrophages. A bigger concern is whether MaudOs are truly an integrated and informative system.

Specific comments:

1. The imaging and migration studies provide strong evidence that MaudOs are truly integrating macrophages into organoids. Still, there is some missing information that would be helpful. Are the macrophages randomly distributed or are they enriched in and around organoids? Does this change when the organoids are differentiated?
2. The authors show that MaudOs have a distinct transcriptional program when compared with independently cultured macrophages and organoids. They use this result to conclude that MaudOs are functionally distinct from its two constituents. Is this a fair comparison? It would be much more compelling if they performed RNA-seq after separating organoids and macrophages from MaudOs. scRNA-seq may be better. Alternatively, they can show by microscopy that cell types make different proteins in the co-culture compared with mono-culture.
3. I also found it difficult to evaluate whether the immune responses were specific to MaudOs or the simple additive response of macrophages and organoids. In other words, did we really learn something new from MaudOs or could we have arrived at the same conclusion by studying macrophages and organoids separately? As an example, Fig 1G shows that the IL-6, IL-1b, and TNF response to LPS by MaudOs can be explained completely by macrophages. In Fig 4M-O, the authors show that the effect of acetate on organoids is canceled out by its effect on macrophages when examining OC43 viral replication in MaudOs. Are there clear examples in their system where there is synergy or a completely unexpected response from MaudOs that is not predicted based on the mono-cultures?
4. A more quantitative comparison between MaudOs and mono-cultures could be helpful. Are the cell counts and density similar in the co-culture and mono-cultures? Could the Matrigel or other culturing conditions be altering macrophage responses (and vice versa)?

Reviewer #3

(Remarks to the Author)

Reviewer #4

(Remarks to the Author)

Here, the authors address a notable limitation of organoids, the lack of an immune system, by integrating intestinal organoids with human macrophages, termed MaudOs. Using MaudOs, the authors investigate antiviral responses to enteric infections and the subsequent macrophage-mediated inflammatory response. MaudOs infected with echoviruses showed antiviral responses but limited inflammatory response, while rotavirus and OC43 infection induced both antiviral and inflammatory responses, matching the pathology seen in human infection. They propose treatment of enteric infections by combinations of antivirals and anti-inflammatory treatments to both reduce viral burden and mitigate any deleterious effects of excessive mucosal inflammation. As a proof of concept, they demonstrate the combination of nirmatrelvir and acetate or budesonide significantly decreased both viral replication and inflammation in MaudOs.

(1) Within MaudOs, it is unclear if the macrophages are in direct contact with the organoid epithelium, which would be important to modeling the intestinal host response. Most images of MaudOs reported are widefield, or appear to be max projections, making the z position of macrophages compared to the organoid difficult to distinguish. Additionally, the H&E image provided is of low quality, making it difficult to identify macrophages among epithelial cells shed from the organoids.

(2) The authors should provide further information on their MaudO infection model. Could infectious virus be recovered from MaudO viral infections? Additionally, did these infections induce any significant cell death?

(3) The authors should clarify if any differentiation of epithelial cells within MaudOs was detected when treated with acetate alone or acetate and OC43, as they report differentiated organoids integrated into MaudOs caused dramatic differences in viral susceptibility. Within the gut, epithelial cells are likely constitutively exposed to acetate, and maturation of organoids in response to commensal bacteria or metabolites has been reported.

(4) Some relevant detail is not included in the methods. Authors should state what R packages were used for RNA sequencing pathway analysis. Authors should also mention where in the small intestine the biopsies for organoid generation

were procured from, as this can alter viral susceptibility.

Reviewer #5

(Remarks to the Author)

Version 1:

Reviewer comments:

Reviewer #2

(Remarks to the Author)

The authors have added several new experiments and analysis to improve the paper. I am not completely convinced by the response to Reviewer #2 Critique #4. The strongest evidence the authors present in the manuscript is that there are unique gene expression patterns in the co-culture. However, I acknowledge the effort the authors put into the revision and that extensive future studies will be necessary to understand the functional consequences of this gene expression pattern and demonstrate that it is physiologically relevant.

Reviewer #3

(Remarks to the Author)

Reviewer #4

(Remarks to the Author)

My comments as reviewer 4 were properly addressed, and I would like to thank the authors for performing so many additional experiments and analyses in addition to an already extensive study.

I was also asked by the editor to review the author responses to comments from reviewer 1.

Reviewer 1, comment 1

The use of organoids from just one donor is limiting. Within the organoid field, inclusion of at least 2-3 donors is typically considered more acceptable. The inclusion of OC43 viral infection and inflammatory analyses for an additional two donors does mostly address this and suggests minimal donor variation. However, these lack the depth of the rest of the experiments. With that said, repeating such an extensive study with additional donors is not necessary nor is it feasible. A statement in the methods or rationale which explains the use of only one donor for most experiments as representative would be sufficient to inform readers of this potential limitation.

Response to Reviewer #1:

Reproducibility, donor variation and virus characteristics.

1. (i) The manuscript contains an extensive amount of data in Fig.s with subFig.s the Methods section lacks information about what these replicates are exactly. ... organoids from the same donor? Different donors? What is the origin of the donors? (ii) Virus infections in organoids may be donor dependent, and host responses are most certainly donor dependent. This issue is not addressed in the manuscript.

Response: (i) We initially established and optimized the MavgOs model using three organoid lines derived from intestinal biopsies of different donors. Once the model was optimized, MavgOs incorporating organoids from donor#1 were used for most experiments as a representative system. We have now included images of MavgOs before and after optimization using these three different organoid lines (Figure 1C). In our study, biological replicates refer to independent organoid or MavgOs cultures, each derived from a separate well. Details regarding the origin of organoids, ethics approval, and replicates have now been added to the Methods section.

(ii) Indeed, viral infections can result in varying clinical manifestations among patients, and we agree that organoid-based infections may be donor-dependent. In the previous version, we tested viral infections in a single MaugOs (donor#1) as a representative. To address this concern, we now further performed OC43 virus infections in MaugOs

incorporating organoids from two additional donors. As shown in Figure 2R, both MaugOs models supported OC43 virus infections. Intracellular viral copy numbers appeared slightly higher in MaugOs-donor#3 than in MaugOs-donor#2 at 36 hours post-infection, although this was not statistically significant. Both MaugOs exhibited robust antiviral and inflammatory responses, with MaugOs-donor#3 displaying higher expression of certain inflammation- and antiviral-associated genes, such as IL10, IFIT1 and STAT1, compared to MaugOs-donor#2 (Figure 2S). Overall, our findings indicate that OC43 infection in MaugOs integrating organoids from different donors follows a similar pattern, though the infection level and host response varied slightly. These new results have been incorporated into the revised manuscript.

2. The justification for the choice of viruses is unsatisfactory (there are many enteroviruses, why echovirus 1 and 6? Are these the main gastro-intestinal enteroviruses? Or are they the most common? Or the most related to serious disease?). .. what strains and/or accession numbers, are these clinical isolates or lab-adapted strains? How were these viruses cultured before infecting the organoids?

Response: Indeed, echovirus is a large group of viruses associated with enteric disease in humans. We selected these two types mainly because, 1) like other enteroviruses that primarily infect by the fecal–oral route and target the gastrointestinal epithelium early during their life cycles (we have added this justification in the manuscript). They are associated with gastrointestinal symptoms at least in some patients, but the exact prevalence remains to be further investigated through large epidemiological studies; 2) and practically we have two strains in our lab. We now have provided the details of viral strains in the Methods section and Supplementary Table 4.

3. Fig 2: the relative viral RNA levels, relative to what? How is this calculated?

Response: Apologize for the confusion. For the quantification of viral RNA levels in Figure 2, we selected viral RNA levels (Gapdh as reference gene) at 1 hpi as baseline, and the viral RNA levels at 12, 24, and 36 hpi were expressed relative to the RNA levels at 1 hpi. This now has been clarified in the legend of Figure 2.

4. Fig S3: here the read out is log viral copies /mg RNA, with very high copies already at day 0, was this also seen in the other replication experiments?

Response: This is primarily due to the presence of non-infectious viruses in the inoculum (common in many

viruses), and the (defective) viral genomes can still be detected by qRT-PCR, which masks the actually increased level of viral replication. This is very clear when assessing the

infectious virus titers. In our study, the initial infectious titers at day 0 (1 h post-inoculation) were either undetectable or below 2 log₁₀, which is quite low. However, at 36 h post-infection the infectious titers can reach to nearly 4 log₁₀ (Fig. 2C,D,I,N).

Physiological Relevance of Models.

5. The authors discuss the physiological relevance of their model and critique the lack of heterogeneity in current models. However, the methodsdoes not fully recapitulate the heterogeneity of the human intestinal environment. A more physiologically representative model, incorporating multiple cell types simultaneously, could provide deeper insights ... For instance, the activation of PPAR γ , a pathway implicated in various viral infections, The authors might consider discussing whether such heterogeneity would enhance the translational value of their model.

Response: We agree that cellular heterogeneity is crucial for accurately recapitulating the complexity of the intestinal environment. In our study, the primary intestinal organoids inherently comprise multiple cell types, including stem cells, enterocytes, goblet cells, and enteroendocrine cells, providing the MaugOs model with greater heterogeneity than conventional cell lines (Sato et al., 2009. *Nature* **459**, 262–265). However, we acknowledge that the diversity and abundance of cell types in MaugOs remain limited and do not fully capture the complexity of the *in vivo* intestinal environment. We highly agree with this reviewer’s suggestions that incorporating additional cell types could enhance the translational relevance of our model. We now have further discussed this point and outline future directions for improving our model.

Novelty of Findings.

6. co-cultures of gut organoid models with macrophages have been described .. (already in 2017 by Noell at all, Scientific Reports). References to these studies are lacking and should be included.

Response: Thank you for pointing out the previous work by Noell et al. We have now discussed this study and incorporated the citation in the revised manuscript.

7. The authors highlight their model’s ability to elucidate cellular tropism for coronaviruses, emphasizing enterocytes as target cells. However, this observation is already well-established in literature. If the goal was to validate the model for studying cellular tropism, this should be explicitly stated. Otherwise, the current

framing may imply novelty where it is not warranted.

Response: Actually, the integration of cell type-differentiated organoids into MaugOs was aiming to enhance the versatility of our model for studying viral infections, with OC43 virus used as an example. We observed that OC43 preferentially targeted enterocyte-differentiated organoids, highlighting the potential of MaugOs incorporating differentiated organoids for investigating cellular tropism of enteric viruses. While enterocytes are well-recognized target cells for coronaviruses SARS-CoV-2 (we didn't find such report on OC43), our findings highlight the broader applicability of this model in dissecting cell-type specificity across different enteric viruses. We have now further discussed the potential of MaugOs in studying viral tropism in the revised manuscript.

Compound Selection and Dosing Information.

8. In Figs 4 and 6, the authors mention the compounds used (e.g., acetate, nirmatrelvir, and budesonide) but only provide concentration details for a subset of these. Additional information about origin (company where purchased), what solution buffer is used to dissolve the compounds, dosages, treatment frequency, and duration for all compounds would improve the reproducibility and interpretation of the results.

Response: As suggested, we have included a supplementary table with the details of all reagents and antibodies used in our study. Additionally, we now described the dosages, treatment frequency and duration for all compounds in the Materials and Methods section.

9. The data in Fig 4 and 6 are not convincing as they lack the control of the dissolvent buffer. DMSO or ethanol-based buffers could have an effect on the cells, therefore it should be proven that the effects the authors claim are in fact the effect of the SCFA or the antiviral compounds and not just due to the composition of the buffer.

Response: Figure 4 and 6 of the previous version is now Figure 5 and 7 in the updated version. For the experiments involving SCFAs, we diluted SCFAs in the culture medium. The control group was treated with culture medium only, ensuring that any observed effects were attributable to SCFAs. We have revised the text to emphasize this point.

For nirmatrelvir and budesonide, these were dissolved in DMSO. We now have conducted additional experiments to confirm that DMSO at the concentrations used in our study does not significantly affect viral replication. These validation results have been included in the Supplementary Fig. 7A.

10. Furthermore, a discussion of how the concentrations of the antivirals relate to clinically achievable dosages would be valuable, as clinical translatability is critical in such studies

11. The rationale behind the selection of compounds is not fully discussed. Were these compounds chosen as proof-of-concept agents for model validation, or are they believed to have clinical relevance? If the latter, the authors should discuss how these compounds could be implemented in clinical settings and whether their model could provide insights into their safety and efficacy. Addressing these questions would enhance the manuscript's translational significance.

Response: Here we address these two interconnected question together. In our study, we aim to provide a proof-of-concept of developing treatments using MaugOs model, and clinically relevant concentration is our first option. For treating coronavirus infections, ours and other studies have extensively demonstrated that nirmatrelvir, the key component of PAXLOVID (FDA-approved for COVID-19), can potently inhibit coronaviruses including OC43 and SARS-CoV-2. We thus selected nirmatrelvir and used a clinically relevant concentration of 1 μ M, which was effective against coronavirus infection and did not exhibit cytotoxicity in the MaugOs model. For anti-inflammatory treatments, we specifically screened 10 clinically used immunosuppressants. And the plasma Cmax of many suppressants, for example Dexamethasone, can reach to 1-2 μ M in patients (e.g., receiving 60 mg/day) (Jobe et al., 2019. *Clin Transl Sci*), we thus selected 1 μ M concentration for inhibiting inflammatory response in MaugOs model. Overall, we aim to provide a proof-of-concept of developing treatments in MaugOs and clinical relevant dosages were selected to increase the translational value . As suggested, we have now discussed the importance of clinically achievable dosage in the revised manuscript.

Minor Comments

1. Ethical statement. The authors do not report on the ethics and ethical considerations in the manuscript and this should be addressed.

Response: The use of human intestinal tissue for research purpose in this study was approved by the Medical Ethical Council of the Erasmus MC, and informed consent was given (MEC-2021-0432; MEC-2023-0629). We now have provided this information in the Methods.

2. Macrophage Characterization.

In the methods section, the authors state that macrophages were differentiated and characterized using monocyte and macrophage markers but do not specify which markers were used or their dilutions. Providing these details would enhance reproducibility.

Response: In the previous version, we examined the expression of CD32 and CD14 in THP-1 monocytes and macrophages (Supplementary Fig.1A). We now further characterized PBMC- and iPSC-derived macrophages using CD16, CD68, CD80 and HLA DR antibodies by flow cytometry analysis, and performed immunofluorescence staining by anti-CD68 antibody (Supplementary Fig. 1O, P). The details of these characterizations now have been described in the Methods section.

3. Reference Gene Stability. The authors mention the use of GAPDH as a reference gene in their qPCR analyses. Did the authors evaluate the stability of GAPDH expression in organoids and macrophages following viral infection? Viral infections can alter host gene expression, potentially affecting the suitability of certain reference genes and introducing biases into the data.

Response: Using our transcriptomic data, we further compared the expression level of GAPDH gene in macrophages, organoids, and MavgOs with and without viral infections. As shown in the right figure (not included in the manuscript), GAPDH expression remains stable across macrophages, organoids, and MavgOs after viral infection. We thus believe GAPDH is qualified as the reference gene in our study.

4. Origin of antibodies. There is no mentioning of the origin of the antibodies used for staining in the Method section. Such information would help the reproducibility of the data by other groups.

Response: As suggested, we now have provided the details of antibodies, ELISA kits, drug compounds and other reagents in the Supplementary Table 4.

5. Fig. S1F (Viability Data). Could the authors clarify the y-axis of this graph? Providing the units and explaining how the data should be interpreted would improve clarity.

Response: The Figure S1F in the previous version now is Figure S1G in the updated version. The y-axis of Fig. S1G represents the raw readout value of fluorescence units (λ_{Exc} 530nm/ λ_{Em} 590nm) from CytoFluor Series 4000 machine. We now included this information in the updated supplementary Figure 1G and further described in the Methods.

6. Fig. S1I (IL-33 Quantification). The authors state that IL-33 was quantified, but this is not shown in Fig. S1I. Instead, TNF α is displayed. Please clarify this inconsistency.

Response: We apologize for causing confusion. The quantification shown in Figure S1I (now supplementary Figure 1K) corresponds to TNF α , not IL-33. We now have corrected this in the revised manuscript.

7. References. The reviewer noticed that the referencing is not always accurate. (i). For example, ref 4 is a very specific study about characteristics of echovirus 1, while the authors make a statement about the enterovirus as a large group that affects many people worldwide. (ii). Furthermore, it strikes me that there seems to be a substantial amount (>10%) of referring to the work of the authors (auto-referencing), also for general statements (for example, line 55 ref 8).

Response: We apologize for causing confusion. In the revised manuscript, we reduced the self-citations, and we have thoroughly reviewed all citations to ensure they are appropriately placed and accurately support the corresponding statements.

Responses to reviewer #2:

Major Comments

1. The cross comparison between different stimuli and viruses was much appreciated and interesting. I also agree with the conclusion..... I see tremendous value in developing the type of approach featured in this current study..... A bigger concern is whether MavgOs are truly an integrated and informative system.

2. The imaging and migration studies provide strong evidence that MavgOs are truly integrating macrophages into organoids. Still, there is some missing information that would be helpful. Are the macrophages randomly distributed or are they enriched in and around organoids? Does this change when the organoids are differentiated?

Response: Here we address these two interconnected question together.

First, we would like to thank this reviewer's praise to our work. In the revised manuscript, we now provided the immunostaining images of MavgOs using intestinal organoids derived from three different donors. Compared to the morphology of pre-optimized MavgOs, we showed that MavgOs were physically integrated by optimized protocol (Figure 1C). To further visualize the active integrating process of macrophages and organoid cells, we now performed time-lapse videos. As shown in the video and captured representative images (Supplementary movies 1; Figure 1B), we observed that macrophages and disrupted organoids were able to gradually migrate and integrate to form MavgOs. We also captured additional images for MavgOs integrating differentiated

organoids. As shown in Supplementary Fig.4A, macrophages (green color) surrounded organoids and some embedded into organoids. Overall, these results demonstrated that macrophages were not randomly distributed but tightly surround and enriched in organoids.

Figure 1
New data

3. The authors show that MavgOs have a distinct transcriptional program when compared with independently cultured macrophages and organoids. They use this result to conclude that MavgOs are functionally distinct from its two constituents. Is this a fair comparison? It would be much more compelling if they performed RNA-seq after separating organoids and macrophages from MavgOs. scRNA-seq may be

better. Alternatively, they can show by microscopy that cell types make different proteins in the co-culture compared with mono-culture.

Response: In our study, genome-wide transcriptomic analysis identified 92 genes exclusively expressed in MavgOs, which indicated the active interactions between macrophages and organoids (Figure 1E). To further consolidate this finding, we now performed additional analysis using our bulk-RNA sequencing data. By comparing viral infections in MavgOs, organoids and macrophages, we observed 127 genes uniquely expressed in MavgOs by EV1 infection. Similar gene expression pattern was also observed in MavgOs with OC43 infection

(Supplementary Figure 3C). Consistently, differential gene expression analysis showed thousands of differentially expressed genes were uniquely regulated in MavgOs. Overall, these findings indicate that MavgOs were not merely simple addition of organoids and macrophages, it encompasses distinct gene expression pattern and generates unique virus-host infections. We feel it is challenging to perform bulk RNA-seq after separating organoids and macrophages from MavgOs, as this would require enzymatic digestion of MavgOs into single cells to isolate macrophages. Instead, we highly agree that single-cell RNA sequencing would provide a more precise interpretation of the model, particularly when integrating additional immune cell types into MavgOs. While we are not able to perform scRNA-seq in the current study, we anticipate incorporating this approach in the follow-up study.

4. (i). I also found it difficult to evaluate whether the immune responses were specific to MavgOs or the simple additive response of macrophages and organoids. In other words, did we really learn something new from MavgOs or could we have arrived at the same conclusion by studying macrophages and organoids separately? As an example, Fig 1G shows that the IL-6, IL-1b, and TNF response to LPS by MavgOs can be explained completely by macrophages. In Fig 4M-O, the authors show that the effect of acetate on organoids is canceled out by its effect on macrophages when examining OC43 viral replication in MavgOs. Are there clear examples in their system where there is synergy or a completely unexpected response from MavgOs that is not predicted based on the mono-cultures?

Response: To our experience so far, the successful establishment of MavgOs is mainly attributed to fine-tuning the physical properties of the system rather than its biological components (Fig. 1B and C, see figure in question 1-2 of Reviewer 1). Consequently, the inflammatory responses observed in MavgOs are largely driven by macrophages. However, we also observed active macrophage-organoid interactions (as explained in above question; Supplementary Figure 3C and D), indicating that MavgOs represent more than a simple additive system of these two cell types.

Actually, the MavgOs model has several distinct advantages over macrophages or organoids alone in studying enteric infections. First, it more accurately recapitulates the course of enteric viral infections in the intestinal tract. By first inoculating organoids with the virus and then integrating macrophages to assemble MavgOs, we simulate the clinical scenario in which viruses initially infect and penetrate the intestinal epithelium before disseminating to macrophages. Second, MavgOs enable the simultaneous investigation of both antiviral and inflammatory responses to viral infections. Third, MavgOs encompass greater translational value than organoids or macrophages alone. With MavgOs, we can develop combination therapies for simultaneously inhibiting viral replication and inflammatory response.

Thus, MavgOs were designed to provide a more physiologically relevant system for investigating enteric viral infections and advancing therapeutic strategies, rather than studying organoids or macrophages itself.

5. A more quantitative comparison between MavgOs and mono-cultures could be helpful. Are the cell counts and density similar in the co-culture and mono-cultures? Could the Matrigel or other culturing conditions be altering macrophage responses (and vice versa)?

Response: During the optimization of MavgOs, we tested different number of macrophages and different Matrigel concentration for establishing MavgOs. Our results showed that integrating 10^5 macrophages generally displayed better inflammatory responses (Supplementary Fig. 1B). When comparing co-culture and mono-cultures, we ensured that the same cell numbers were used across conditions.

Regarding the Matrigel, we tested three different concentrations of Matrigel. We found that 5% Matrigel concentration cannot support a well-3D morphology (Supplementary Fig. 1C). We then compared 10% and 20% concentration of Matrigel for their influence to inflammatory responses in MavgOs. As shown in Supplementary Fig. 1D, 20% Matrigel largely inhibited key inflammatory gene expression by LPS stimulation, we thus used 10% concentration Matrigel as the optimal support matrix for our model. These details have now been included in the supplementary file.

Responses to reviewer #3:

I co-reviewed this manuscript with one of the reviewers who provided the listed reports. This is part of the Nature Communications initiative to facilitate training in peer review and to provide appropriate recognition for Early Career Researchers who co-review manuscripts.

Response: Thank you for your time and efforts in evaluating our manuscript.

Responses to reviewer #4:

1. Within MaugOs, it is unclear if the macrophages are in direct contact with the organoid epithelium, which would be important to modeling the intestinal host response. Most images of MaugOs reported are widefield, or appear to be max projections, making the z position of macrophages compared to the organoid difficult to distinguish. Additionally, the H&E image provided is of low quality, making it difficult to identify macrophages among epithelial cells shed from the organoids.

Response: To further clarify the spatial relationship between macrophages and organoids within MaugOs, we now have performed time-lapse video which captured the integrating process of macrophages and organoids (Figure 1B; Supplementary movies 1, picture in question 1-2 of Reviewer 1). We also provided the morphological images of MaugOs before and after optimization (Figure 1C; picture in question 1-2 of Reviewer 1). Furthermore, we now performed a Z-stack visualization of MaugOs, which showed macrophages incorporated into multiple layers of organoids structures (Supplementary Fig.1I; Supplementary movies 2). These results collectively confirmed that macrophages are directly contact with organoids within MaugOs. As suggested, we have removed the H&E image in Figure 1. Instead, we now performed IHC staining, which clearly demonstrates the integration of macrophages with organoids in the MaugOs model (Supplementary Fig.1H).

2. The authors should provide further information on their MaugO infection model. (i) Could infectious virus be recovered from MaugO viral infections? (ii) Additionally, did these infections induce any significant cell death?

Response: (i). To demonstrate whether the produced viruses by MaugOs are infectious, we now further performed TCID₅₀ assay for the supernatant from infected MaugOs. Our results showed that the virus infectious titers in the supernatant were significantly increased at 36 hours compared to 1 hour post-inoculation (Figure 2C, D, I, N). These findings demonstrated that the produced viruses from MaugOs model were highly infectious.

(ii). To investigate the potential cell death induced by virus infections in MaugOs, we first analyzed our bulk-RNA sequencing data. We found that both EV1 and OC43 virus infection significantly enriched the geneset of apoptosis signaling pathway (Figure 3D, see below figure). We then further performed propidium iodide (PI; stain dead cells) and Hoechst (stain cell nucleus) staining in MaugOs following EV1 and OC43 infections. Importantly, we observed many PI-positive cells in

staining in MaugOs following EV1 and OC43 infections. Importantly, we observed many PI-positive cells in

EV1 or OC43 infected MaugOs, indicating the cell death induced by virus infections (Figure 3E). We further measured lactate dehydrogenase (LDH), a classical indicator of cellular damage. Consistently, both EV1 and OC43 triggered significant LDH release in MaugOs (Figure 3F). We now have included these new data in the revised manuscript.

3. The authors should clarify if any differentiation of epithelial cells within MaugOs was detected when treated with acetate alone or acetate and OC43, as they report differentiated organoids integrated into MaugOs caused dramatic differences in viral susceptibility. Within the gut, epithelial cells are likely constitutively exposed to acetate, and maturation of organoids in response to commensal bacteria or metabolites has been reported.

Response: Indeed, gut metabolites have been reported to influence the maturation of organoids. Since acetate can significantly upregulate RNA expression of enterocyte marker Villin in intestinal organoids (Supplementary Fig.5F), we now further tested the expression of Villin in MaugOs by qRT-PCR. We found that both acetate treatment and OC43 virus infection were able to upregulate the RNA level of Villin in MaugOs, and their combination further augmented Villin expression (Supplementary Fig. 5G). We now described these results in the revised manuscript.

4. Some relevant detail is not included in the methods. Authors should state what R packages were used for RNA sequencing pathway analysis. Authors should also mention where in the small intestine the biopsies for organoid generation were procured from, as this can alter viral susceptibility.

Response: In the revised manuscript, we now included a detailed table that listed all reagents, antibodies, primers, viruses, and other materials used in the study (Supplementary Table 4). We also provided the details of organoids origin and ethical approval in the *Methods* section. For the bioinformatics analysis of transcriptomic data, we now described the R packages we used in the Methods.

Responses to reviewer #5:

Concerns: I co-reviewed this manuscript with one of the reviewers who provided the listed reports. This is part of the Nature Communications initiative to facilitate training in peer review and to provide appropriate recognition for Early Career Researchers who co-review manuscripts.

Response: Thank you for your time and efforts in evaluating our manuscript.

General remarks:

We would like to thank all the reviewers for their constructive suggestions. We now have revised the manuscript according to the comments of the editorial team and reviewers. The corresponding changes in text and legends were highlighted in RED. In addition, we have reviewed grammar, (Nature Communications) style, structure and punctuation throughout the manuscript, but these minor changes were not highlighted.

Reviewer 1

The use of organoids from just one donor is limiting. Within the organoid field, inclusion of at least 2-3 donors is typically considered more acceptable. The inclusion of OC43 viral infection and inflammatory analyses for an additional two donors does mostly address this and suggests minimal donor variation. However, these lack the depth of the rest of the experiments. With that said, repeating such an extensive study with additional donors is not necessary nor is it feasible. A statement in the methods or rationale which explains the use of only one donor for most experiments as representative would be sufficient to inform readers of this potential limitation

Response to reviewer 1: As suggested, we have now provided a statement “For the characterization and application of the MauqOs model, organoids derived from three different donors were used to validate key findings, while one representative donor was employed for the majority of experiments in this study” in the Methods section.

Reviewer #2:

The authors have added several new experiments and analysis to improve the paper. I am not completely convinced by the response to Reviewer #2 Critique #4. The strongest evidence the authors present in the manuscript is that there are unique gene expression patterns in the co-culture. However, I acknowledge the effort the authors put into the revision and that extensive future studies will be necessary to understand the functional consequences of this gene expression pattern and demonstrate that it is physiologically relevant.

Reviewer #3:

I co-reviewed this manuscript with one of the reviewers who provided the listed

reports. This is part of the Nature Communications initiative to facilitate training in peer review and to provide appropriate recognition for Early Career Researchers who co-review manuscripts.

Reviewer #4:

My comments as reviewer 4 were properly addressed, and I would like to thank the authors for performing so many additional experiments and analyses in addition to an already extensive study.

Response to all the reviewers: We thank the reviewers for strictly reviewing our study, which has substantially improved our manuscript.